# Multi-channel learning for integrating structural hierarchies into context-dependent molecular representation

Yue Wan[1], Jialu Wu[2], Tingjun Hou ®[2] ✉, Chang-Yu Hsieh ®[2] ✉ & Xiaowei Jia ®[1] ✉

Reliable molecular property prediction is essential for various scientific endeavors and industrial applications, such as drug discovery. However, the data scarcity, combined with the highly non-linear causal relationships between physicochemical and biological properties and conventional molecular featurization schemes, complicates the development of robust molecular machine learning models. Self-supervised learning (SSL) has emerged as a popular solution, utilizing large-scale, unannotated molecular data to learn a foundational representation of chemical space that might be advantageous for downstream tasks. Yet, existing molecular SSL methods largely overlook chemical knowledge, including molecular structure similarity, scaffold composition, and the context-dependent aspects of molecular properties when operating over the chemical space. They also struggle to learn the subtle variations in structure-activity relationship. This paper introduces a multi-channel pre-training framework that learns robust and generalizable chemical knowledge. It leverages the structural hierarchy within the molecule, embeds them through distinct pre-training tasks across channels, and aggregates channel information in a task-specific manner during fine-tuning. Our approach demonstrates competitive performance across various molecular property benchmarks and offers strong advantages in particularly challenging yet ubiquitous scenarios like activity cliffs.

Empowered by the advancement of machine learning techniques, molecular machine learning has shown its great potential in computational chemistry and drug discovery[1,2]. The data-driven protocol allows the model to infer biochemical behaviors from simple representations like SMILES sequence[3] and molecular graph, enabling fast identification of drug candidates via rapid screening of vast chemical spaces[4], as well as prediction of binding affinity, toxicity, and other pharmacological properties[5,6]. These advancements significantly accelerate the drug discovery procedures, saving time and efforts from the traditional wet-lab experiments[7,8]. However, it is fundamentally challenging to learn an effective and robust molecular representation

via machine learning, limited by the expensive gathering of precise biochemical labels and the complexity underlying the structure-property relationships (SPR). With data scarcity, models may become overly adapted to specific structural patterns within the training molecules, making it fail to generalize to the broader chemical space. In addition, the challenge of "activity cliffs"[9] in drug discovery, where minor changes in molecular structure significantly alter the biological activity, further impose obstacles in developing accurate Quantitative SPR (QSPR) models[6,10–12]. Activity cliffs refer to the concept where structurally similar molecules may exhibit significantly different biological activities. Understanding activity cliffs is crucial for

[1]University of Pittsburgh, Department of Computer Science, Pittsburgh, PA 15260, USA. [2]Innovation Institute for Artificial Intelligence in Medicine of Zhejiang University, College of Pharmaceutical Sciences, Zhejiang University, Hangzhou 310058, China. ✉e-mail: tingjunhou@zju.edu.cn; kimhsieh@zju.edu.cn; xiaowei@pitt.edu

drug discovery, as it enables more efficient lead optimization and enhances predictive modeling with better identification and development of potential drug candidates[13,14].

Inspired by the success of the pretrain-finetune workflow in computer vision[15] and natural language processing[16,17], various methods in molecule self-supervised learning (SSL)[18-25] have emerged. In the self-supervised setting, machine learning models are pre-trained to learn generic molecular representations by optimizing the performance on pre-defined tasks on large-scale unannotated molecule data. These tasks are designed in a way such that solving them requires identification of important structural patterns and understanding of rudimentary chemical knowledge. Existing molecule SSL methods can be mainly classified into two categories: predictive and contrastive. Predictive learning[18-20,26-28] aims to predict structural components given contexts at different levels, which mainly focuses on intra-data relationship. These methods often follow the conventional pipeline of reconstructing the molecular information from masked inputs. Contrastive learning[20,22,23,29-32], initially proposed in computer vision[33], aims to learn the inter-data relationship by pulling semantic-similar data samples closer and pushing semantic-dissimilar samples apart in the representation space. Note that this idea aligns well with common heuristics in chemistry, where structurally similar molecules are likely to exhibit similar physicochemical and biological properties. Most works attempt to generalize the same SSL methods across multiple domains (e.g., social networks and molecular graphs), whereas it is unsure whether the same learning schemes are compatible to all settings. Recently, several studies have pointed out that existing SSL methods may fail to learn effective molecular representation. RePRA[34] was proposed to measure the representations' potential in solving activity cliffs and scaffold hopping[35]. These are challenging yet ubiquitous tasks in drug discovery that requires the model to understand subtle chemical knowledge behind SPR. Experiments showed that most pre-trained representations perform worse than molecular fingerprints. The work in[36] explored various molecular SSL methods and observed that some pre-training strategies can only bring marginal improvement, while some may induce negative transfer. Meanwhile, studies have also highlighted the incompetence of several pre-trained representations in predicting binding potency under activity cliffs[6,12,37].

This work aims to enhance molecular representation learning that encodes robust and generalizable chemical knowledge. We start by identifying the two major drawbacks in existing methods: Firstly, in contrastive learning, the conventional formulations of the semantic-similar/dissimilar (i.e., positive/negative) samples are not well-tailored for molecular graphs. Most graph contrastive methods generate positive samples via graph perturbation, such as node/edge addition/deletion[20,21,21-23]. However, when applied on molecular graphs, chemical validity may be easily challenged. Molecules may also lose essential characteristics by perturbing important motifs (e.g., breaking an aromatic ring), shifting the "semantics" distant away. The negative samples (i.e., different molecules) are often treated equally, which essentially neglects the molecule structural relationship and the presence of specific molecular components; Second, almost all existing works attempt to learn a context-independent molecular representation space, aiming to generalize to various applications. However, this contradicts the fact that molecular properties are often context-dependent, from both the physical (e.g., surrounding environments) and biological (e.g., interaction with proteins) perspectives. In other words, it remains uncertain whether the same SSL tasks could align well with diverse downstream tasks of distinct properties in fine-tuning, thereby leading to the learning gap.

To approach the aforementioned challenges, we introduce a prompt-guided multi-channel learning framework for molecular representation learning. Each of the $k$ channels, guided by a specific prompt token, is responsible for learning one dedicated SSL task. Essentially, the pre-trained model is able to learn $k$ distinct representation spaces. During fine-tuning, a prompt selection module aggregates $k$ representations into a composite representation and uses it for the downstream molecular property predictions. This involves determining which information channel is most relevant to the current application, thereby making the representation context-dependent. We later show how this composite formulation is more resilient to label overfitting and manifests better robustness. In addition, we design the pre-train tasks to form an interpolation from a global view to a local view of the molecular structures. Besides leveraging the global molecule contrastive learning and the local context prediction[19], we introduce the task of scaffold contrastive distancing, highlighting the fundamental role of scaffolds in affecting molecular characteristics and behaviors. Since scaffolds are often treated as starting points for new compound design, scaffold distancing aims to map molecules with similar scaffolds (generated via *scaffold-invariant perturbations*) closer in the representation space. Additionally, it pushes molecules with different scaffolds apart, where the distance margin is computed adaptively based on structure composition difference. Note that scaffold distancing tackles the partial but core view of the molecules. The overall framework is pre-trained using ZINC15[38], and evaluated on 7 molecular property prediction tasks in MoleculeNet[5] and 30 binding potency prediction tasks in MoleculeACE[6]. Learning to leverage information from different channels for different applications, our method surpasses various representation learning baselines in both benchmarks. More importantly, our method is shown to handle the challenge of activity cliffs more effectively, whereas competing approaches are more susceptible to the negative transfer, leading to a substantial performance decline. This suggests that these methods may rely more on surface-level patterns even after pre-training or are more susceptible to knowledge forgetting during fine-tuning, causing them to struggle with challenging problems that require a nuanced understanding of chemical knowledge. On the contrary, our learned representation demonstrates enhanced ability in preserving pre-trained knowledge during fine-tuning, offering improved transferability and robustness compared to other baselines. Case study shows that our method has the potential to identify crucial patterns that contribute to activity cliffs, even when relying solely on topological information.

## Results

The proposed framework (Fig. 1) comprises three major components that differ from the conventional pretrain-finetune paradigm on molecules: (1) The prompt-guided multi-channel learning, (2) contrastive learning with adaptive margin, and (3) scaffold-invariant molecule perturbation. It demonstrates effectiveness on both the molecular property prediction[5] and binding potency prediction[6] benchmarks, offering enhanced robustness and interpretability.

### Prompt-guided multi-channel learning

We introduce a prompt-guided multi-channel learning framework for molecular representation learning, as shown in Fig. 1a. Essentially, the molecular graphs will first go through a unified encoder module, and then diverge into $k$ different channels, each of which is responsible for learning distinct SSL tasks. For each channel, a prompt token $p_i$ is utilized to distinguish levels of molecule representation. This is realized via a prompt-guided readout operation[39], which aggregates atom representations conditionally into molecule representation given the prompt token (Fig. 1e). Our experiment involves three learning channels, which are molecule distancing, scaffold distancing, and context prediction. Each channel focuses on a unique aspect of the molecular structure, enabling molecular representation learning from a set of hierarchical viewpoints, from a global view (i.e., entire molecule), a partial view (i.e., core structure), and down to a local view (i.e., functional groups).

We now briefly introduce the three learning channels. (i) Molecule distancing (Fig. 1b) is achieved using a variant of triplet contrastive

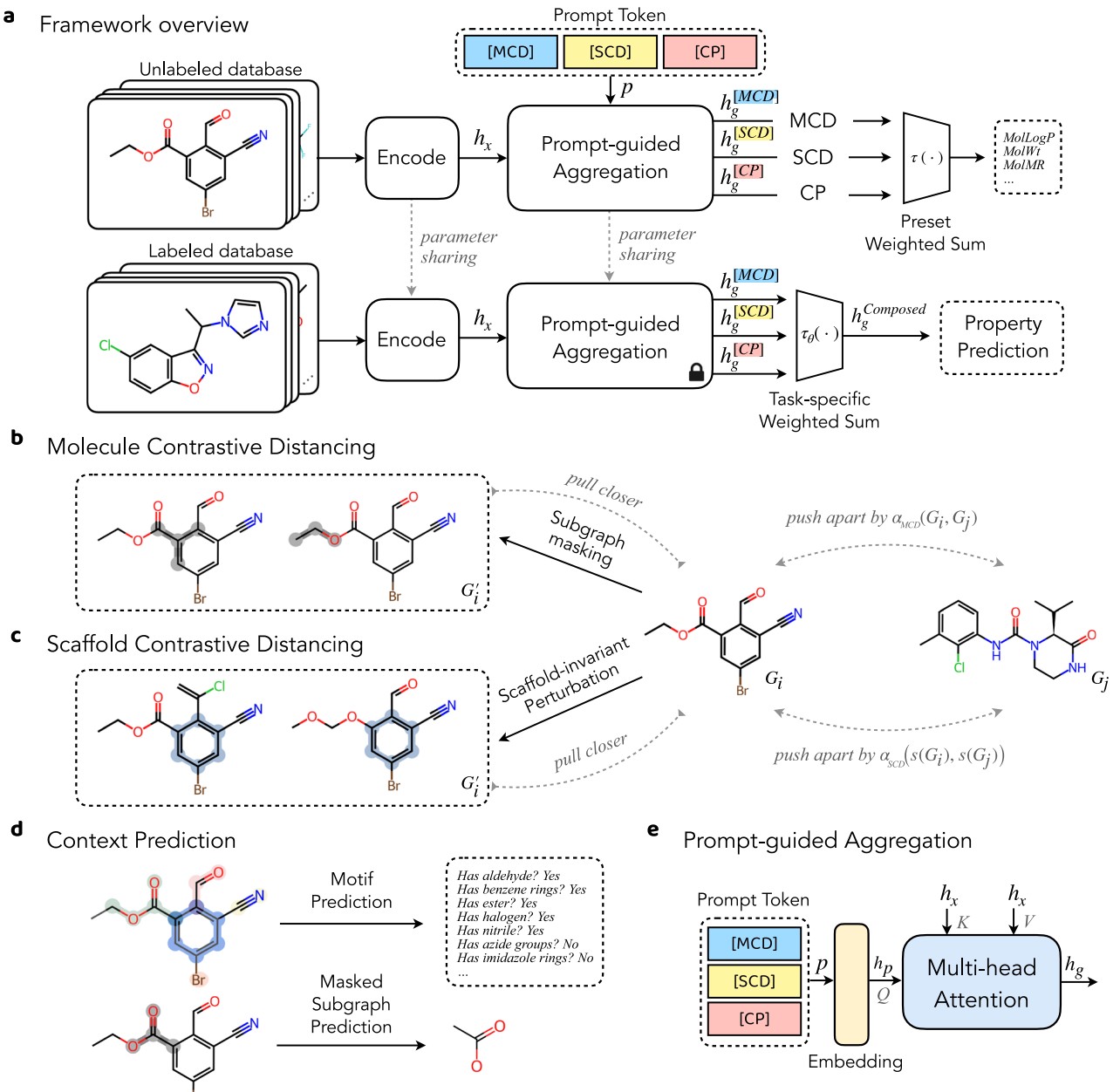

**Fig. 1 | Framework overview. a** The prompt guided pretrain-finetune framework. For each downstream task, the model is optimized additionally on the prompt weight selection, locating the best pre-trained channel compatible with the current application. **b** Molecule contrastive learning (MCD), where the positive samples $G_i'$ from subgraph masking is contrasted against negative samples $G_j$ by an adaptive margin. **c** Scaffold contrastive distancing (SCD), where the positive samples $G_i'$ from scaffold-invariant perturbation is contrasted by against negative samples $G_j$ by an adaptive margin. **d** Context prediction (CP) channel consists of masked subgraph prediction and motif prediction tasks. **e** Prompt-guided aggregation module, which conditionally aggregates atom representations into molecule representation by prompt token. It is realized via a multi-head attention with prompt embedding $h_p$ being the query.

loss[40]. It considers triplet of molecule samples {anchor, positive, negative}, where negative (i.e., dissimilar) samples are pushed apart against the anchor and positive (i.e., similar) samples by a margin $\alpha$. On top of this, we propose the *adaptive margin* $\alpha(\cdot)$, as detailed below. It introduces another level of distancing constraints based on the structural similarity of molecule composition. We follow the work of Molecular Contrastive Learning of Representations (MolCLR)[21] and generate positive samples via molecule subgraph masking. (ii) Scaffold distancing (Fig. 1c) is a contrastive learning task that focuses on scaffold differences. Molecule scaffolds are viewed as the foundation for a range of biologically active molecules. They play a crucial role in drug discovery and medicinal chemistry by providing a starting point for

compound designs with desired pharmacological properties. In other words, molecules with similar scaffolds are more likely to possess similar physical (e.g., solubility, lipophilicity) and biological (e.g., conformational property when interacting with a protein) characteristics, and thereby sharing similar semantics. Scaffold distancing contrasts the *scaffold-invariant molecule perturbations*, as detailed below, against molecules with different scaffolds using the *adaptive margin loss*. (iii) Context prediction (Fig. 1d) involves masked subgraph prediction and motif prediction, which are also adopted in GROVER[19]. For each molecular graph, a random subgraph (i.e., a center atom and its one-hop neighbors) is masked out, and the model aims to reconstruct the subgraph based on its surrounding structures. Motif prediction

aims to predict the existence of functional groups within the molecule. This learning channel mainly focuses on the local view of the molecule by identifying the existence of substructure and functional groups, while molecule distancing and scaffold distancing focus more on the global view and partial view, respectively. Each channel has its own readout layer, termed prompt-guided node aggregation (Fig. 1e). Presumably, the distribution of atom importance should be different across channels (Supplementary Figs. 7–9 and Supplementary Note 6). The learned channel-wise representations exhibit a high correspondence to the associated structural features (Supplementary Fig. 1 and Supplementary Note 1) and support hierarchical iterative clustering of the chemical space (Supplementary Fig. 2 and Supplementary Note 2). To further improve the robustness of the learned knowledge under this framework, we incorporate two regularization tricks on the intra-channel node aggregation and the inter-channel alignment. For the molecule and scaffold channels, the aggregation attentions are more encouraged to span across all atoms and scaffold atoms, respectively. Meanwhile, a small set of supervised tasks is utilized to regularize the composite representation under three prompt weight presets, hence improving the channel alignment. A comprehensive ablation study is provided in Supplementary Fig. 10 and Supplementary Note 7.1.

In the fine-tuning stage, the model is initialized with the same molecule encoder and prompt-guided aggregation modules, along with the learned parameters from pre-training. The parameters in the aggregation modules are fixed during fine-tuning as we aim to use it as a pooling layer independent of the downstream applications. In addition, a prompt-tuning module $\tau_\theta(\cdot)$ is introduced to determine which channel is most relevant to the current application. It essentially learns a task-specific prompt distribution. The prompt weights are utilized to linearly combine the channel-wise information into a composite molecule representation, which is then used for the task-specific prediction. We discover that this approach is more effective than simply concatenating the representations (Supplementary Fig. 11 and Supplementary Note 7.2). We initialize the prompt weights by choosing the candidate which leads to the smoothest quantitative structure-property landscape (i.e., lowest roughness index)[41] of the composite representation. More details are provided in the Method section.

## Contrastive learning with adaptive margin

We further introduce the adaptive margin loss, a variant of the triplet loss[40], that supports the contrastive learning in the first two channels (molecule distancing and scaffold distancing). In the conventional triplet loss, the representation distance between the anchor $G_i$ and negative (i.e., semantic-dissimilar) sample $G_j$ needs to be at least by margin $\alpha$ larger than the distance between the anchor $G_i$ and positive (i.e., semantic similar) sample $G_i'$. Note that this margin remains the same for any triplet considered. However, when applied to molecule triplets, it neglects the known structural relationship between molecules (e.g., co-existence of functional groups). To learn a more fine-grained molecule representation space, we propose to adaptively compute the molecule triplet margin based on the Tanimoto similarity between molecule fingerprints. As shown in Fig. 1b and c, the adaptive margin $\alpha_{MCD}(.)$ considers the molecule structural similarity between $G_i$ and $G_j$, while $\alpha_{SCD}(.)$ considers the scaffold structural similarity between $s(G_i)$ and $s(G_j)$. Another issue with the conventional triplet loss is that it imposes no constraint on the representation space beyond the margin. It means that the actual representation distances are not necessarily to be well correlated with the computed margin, even if the margin constraints are fully satisfied. This is further elaborated the example in Supplementary Fig. 3. Therefore, we include a secondary term into the adaptive margin loss by considering the structural relationship among the anchor and different negative samples. Detailed formulation of the adaptive margin loss is included in Method section. With careful consideration of existing structural similarity, the learned representation space would better capture molecule relationships in a fine-grained representation space. A performance drop is observed when the adaptive margin loss is replaced with the conventional margin loss (Supplementary Fig. 10 and Supplementary Note 7.1).

## Scaffold-invariant molecule perturbation

To generate semantic-similar samples (i.e., positive) for scaffold contrastive distancing, we propose to perturb only the terminal side chains of the molecule. In other words, the molecule scaffold (i.e., core structure) is preserved. This is done by first identifying the side chains and then performing fragment replacement based on a candidate fragment pool. To avoid significant alterations in molecule characteristics, we restrict the amount of changes to be fewer than five atoms. For simplicity reason, we consider the Bemis-Murcko framework as the scaffold. Figure 1c shows a sample perturbation with scaffolds highlighted in blue. In this example, the benzene ring is the identified scaffold, and either the carboxylic ester group or the carbonyl group is perturbed by another functional groups. Note that such perturbation is not limited to atom-level or bond-level editing, but also motif-level.

## Molecular property prediction

To demonstrate the effectiveness of our approach, we first evaluate it on seven challenging classification datasets from MoleculeNet[5], which is a large-scale curated benchmark that covers multiple molecular property domains (e.g., physiology, biophysics). The scaffold splits scheme[18] is applied. The performance is evaluated using the ROC-AUC value. Each experimental result is averaged over three different runs following the prior works[19,21,25]. To demonstrate the effectiveness of our framework across different model architectures, we pre-train both a graph neural network GIN[42] and a graph transformer GPS[43], termed Ours$_{GIN}$ and Ours$_{GPS}$, respectively. We compare our methods with twelve competitive molecular representation learning baselines, which cover a wide variety of pre-train SSL techniques, model architectures, and input representations (i.e., sequence, graph, geometry). Specifically, we first consider four contrastive baselines, including GraphLoG[22], D-SLA[23], MolCLR[21], and KANO[24]. These methods also share the commonality of using GNNs to encode molecular graphs. Besides, we include six predictive baselines, including Hu et al.[18], GROVER[19], MoLFormer[28], KPGT[44], GEM[25], and Uni-Mol[27]. MoLFormer serves as a strong baseline for sequence-based models, while GEM and Uni-Mol are strong baselines for 3D geometry models. We further incorporate GraphMVP[45] and ImageMol[46] as two multi-task learning baselines whose pre-training tasks cover both contrastive and predictive learning. Last but not least, we train the GIN and GPS models from scratch. Table 1 shows the performance comparison results. Our methods improve over the no-pretrain setting (i.e., GIN and GPS) by 12.6% and 8% ROC-AUC in average, respectively. When compared to the other SSL methods, our approach reaches state-of-the-art performances on BBBP, Clintox, BACE and SIDER datasets, while remaining highly competitive in the rest of the tasks. Our overall ROC-AUC score is 0.8% higher than the second best method (Uni-Mol). Notably, Uni-Mol leverages 3D geometric information and is an order of magnitude larger (in term of model parameters) than both Ours$_{GIN}$ and Ours$_{GPS}$.

## Binding potency prediction

We consider MoleculeACE[6] as the second evaluation benchmark. It consists of 30 datasets retrieved from ChEMBL[47]. Each dataset contains binding potency measures (e.g., $K_i$ value) of molecules against a macromolecular target. These datasets mainly focus on the structure-property relationships (SPR), where the phenomenon of activity cliffs is amplified. Activity cliffs refer to the cases where small changes in molecular structure significantly alters its biological activity, and understanding them is crucial for optimizing lead compounds and designing new molecules with desired activities. The phenomenon is also counter-intuitive in machine learning, as the machine learning model tends to make similar predictions given similar inputs. We take

**Table 1 | Fine-tuning results on 7 classification tasks in MoleculeNet using the scaffold splits**

| Methods #task | BBBP 1 | Clintox 2 | MUV 17 | HIV 1 | BACE 1 | Tox21 12 | SIDER 27 | Avg. |
|---|---|---|---|---|---|---|---|---|
| GIN[42] | 65.8 ± 4.5 | 58.0 ± 4.4 | 71.8 ± 2.5 | 75.3 ± 1.9 | 70.1 ± 5.4 | 74.0 ± 0.8 | 57.3 ± 1.6 | 67.5 |
| GPS[43] | 64.8 ± 3.0 | 87.2 ± 0.9 | 69.8 ± 3.8 | 73.1 ± 3.3 | 78.0 ± 3.0 | 74.5 ± 0.6 | 60.8 ± 0.6 | 72.6 |
| Hu et al.[18] | 70.8 ± 1.5 | 72.6 ± 1.5 | 81.3 ± 2.1 | 79.9 ± 0.7 | 84.5 ± 0.7 | 78.7 ± 0.4 | 62.7 ± 0.8 | 75.8 |
| GraphLoG[22] | 72.3 ± 0.9 | 74.7 ± 2.2 | 74.2 ± 1.8 | 75.4 ± 0.6 | 82.2 ± 0.9 | 75.1 ± 0.7 | 61.2 ± 1.1 | 73.6 |
| D-SLA[23] | 72.6 ± 0.8 | 80.2 ± 1.5 | 76.6 ± 0.9 | 78.6 ± 0.4 | 83.8 ± 1.0 | 76.8 ± 0.5 | 60.2 ± 1.1 | 75.5 |
| MolCLR[21] | 73.5 ± 0.4 | 90.4 ± 1.7 | 75.5 ± 1.8 | 77.6 ± 3.2 | 83.5 ± 1.8 | 76.7 ± 2.1 | 60.7 ± 5.7 | 76.8 |
| GraphMVP[45] | 72.4 ± 1.6 | 79.1 ± 2.8 | 77.7 ± 0.6 | 77.0 ± 1.2 | 81.2 ± 0.9 | 75.9 ± 0.5 | 63.9 ± 1.2 | 75.3 |
| GROVER[19] | 69.5 ± 0.1 | 76.2 ± 3.7 | 67.3 ± 1.8 | 68.2 ± 1.1 | 81.0 ± 1.4 | 73.5 ± 0.1 | 65.4 ± 0.1 | 71.6 |
| GEM[25] | 71.8 ± 0.6 | 89.7 ± 2.0 | 77.0 ± 1.5 | 78.0 ± 1.4 | 84.9 ± 1.1 | 78.2 ± 0.3 | 67.2 ± 0.6 | 78.1 |
| ImageMol[46] | 73.9 ± 0.2 | 85.1 ± 1.4 | **82.5 ± 0.8** | 79.7 ± 0.2 | 83.9 ± 0.5 | 77.3 ± 0.1 | 66.0 ± 0.1 | 78.3 |
| KANO[24] | 69.9 ± 1.9 | 90.7 ± 2.2 | 74.7 ± 2.0 | 75.7 ± 0.3 | 82.7 ± 0.9 | 75.8 ± 0.5 | 60.2 ± 1.4 | 75.7 |
| KPGT[44] | 71.4 ± 0.7 | 88.8 ± 2.9 | 75.7 ± 1.4 | 77.9 ± 1.2 | 81.8 ± 2.7 | 78.5 ± 0.5 | 64.7 ± 1.0 | 77.0 |
| MoLFormer[28] | 70.9 ± 1.0 | 91.1 ± 0.9 | 80.5 ± 1.5 | 76.7 ± 0.4 | 83.6 ± 1.1 | 77.3 ± 0.4 | 64.9 ± 0.7 | 77.8 |
| Uni-Mol[27] | 72.9 ± 0.6 | 91.9 ± 1.8 | 82.1 ± 1.3 | **80.8 ± 0.3** | 85.7 ± 0.2 | **79.6 ± 0.5** | 65.9 ± 1.3 | 79.8 |
| Ours$_{GIN}$ | **74.1 ± 0.6** | **95.7 ± 1.2** | 81.2 ± 0.5 | 79.8 ± 0.3 | 85.0 ± 1.1 | 77.5 ± 0.3 | 66.7 ± 0.8 | 80.1 |
| Ours$_{GPS}$ | 73.6 ± 0.7 | 95.1 ± 0.5 | 81.5 ± 0.8 | 80.2 ± 0.5 | **86.1 ± 1.3** | 79.0 ± 0.6 | **68.7 ± 0.2** | **80.6** |

Average Receiver Operating Characteristic Area Under the Curve (ROC-AUC) value is reported, along with the score standard deviation (shown by ± ) from three independent runs. The first two rows of GIN and GPS showcase the performance of the backbone models without pre-training. Best performance is shown in bold.

R-squared as the evaluation metric since relative binding potency ranking is more important than absolute prediction errors (e.g., RMSE) in the real world.

Figure 2a shows the performance of thirteen methods on the MoleculeACE benchmark under stratified splits[6] (Supplementary Table 2). The multi-layer perceptron (MLP) is trained using the ECFP4 fingerprint. We also compare the average performance with respect to the model sizes (i.e., number of parameters) in Fig. 2b, as indicated by its x-axis and the size of the dots, and types of input representations (blue = 1D sequence, red = 2D topological graph, and yellow = 3D geometry). In general, Ours$_{GPS}$ ranks in the second place, but with a significantly smaller model size than KPGT. Besides, almost all methods fail to surpass MLP with fingerprints. MolCLR and GraphLoG also show negative transfer compared to the GIN model. It demonstrates the incompetence of existing methods in learning the nuances of chemical knowledge behind the structural-property relationship, regardless of the input representations, pre-training strategies, and the model sizes[6,12]. One of the reasons behind KPGT's strong performance could be attributed to the fact that the molecule fingerprint and descriptors are heavily embedded within the model. In addition, KPGT is pre-trained using the ChEMBL database[47], such that over 99% of the testing molecules in MoleculeACE are already exposed to the model during the pre-training stage. In contrast, our models only have seen less than 5% of the testing molecules during the pre-training stage. The plot also suggests that there are no clear advantages of molecular representation learning using either 1D sequence or 3D geometry, especially when comparing the performance with Uni-Mol and MoLFormer.

We further study the model performance in relation to the presence of activity cliffs in each dataset, which is measured by the roughness index (ROGI)[41] with respect to the ECFP4 fingerprints and potency labels. A smaller ROGI value indicates a smoother structure-property landscape, hence less activity cliffs. Since ROGI is a quantitative SPR (QSPR) metric that analyzes the overall chemical space, it does not account for any distribution shift between chemical spaces. Hence, its correlation with model performance would be easily obscured by any distribution shift among training, validation, and testing sets (Supplementary Fig. 14). For this reason, all experiments in this work that involves QSPR analysis are performed using random

splits, following the work in[41]. As shown in Fig. 2c and Supplementary Table 3, the model performance is negatively correlated with landscape roughness for all methods, which is consistent with the results in[41]. Compared to the MLP with fingerprint and KPGT, our method is shown to be more robust against tasks with rough structure-property landscapes. In addition, the plot also indicates that our method performs better under data scarcity, where the size of dots represents the size of datasets.

**Representation robustness**

To study the robustness of the learned molecular representation, we propose to probe the fine-tuning process and evaluate the shift in the representation space. Essentially, the shift captures how much pre-trained chemical knowledge is distorted during fine-tuning. We examine the representation space of both the training and validation molecule set at five training timestamps. To clarify, the term "learned representation" refers to the numerical embedding used for the final prediction layer. We choose CHEMBL237_Ki, one of the largest binding potency prediction dataset in MoleculeACE, for the downstream target. For fair comparison, this analysis is performed among Ours$_{GIN}$, GraphLoG, and MolCLR, as these methods only differ by the SSL strategies for pre-training, while using the same pre-train dataset, GNN architecture and hyperparameters. In Fig. 3, each column represents a training timestamp, and each row pair represents the visualization of representation space in training and validation sets. The coloring represents the normalized potency labels for illustrative purposes. We also report four additional metrics that capture the representation characteristics along the training process: 1. *Roughness Index* (i.e., ROGI)[41] captures the landscape roughness of molecular property given a representation. 2. *Rand Index*[48] measures the proximity between fingerprint (ECFP4) clustering and representation clustering, serving as a proxy for the amount of structural information encoded within the current representation space. A higher Rand Index indicates a greater amount of structural knowledge being captured by the representation. 3. *Cliff-noncliff Distance Ratio* indicates the generalizability of the representation towards the activity cliffs. In short, cliff matched molecule pairs (MMPs) (i.e., similar molecules with different labels) should be more distant away compared to the non-cliff MMPs (i.e., similar molecules with similar labels). It is calculated as the ratio of the

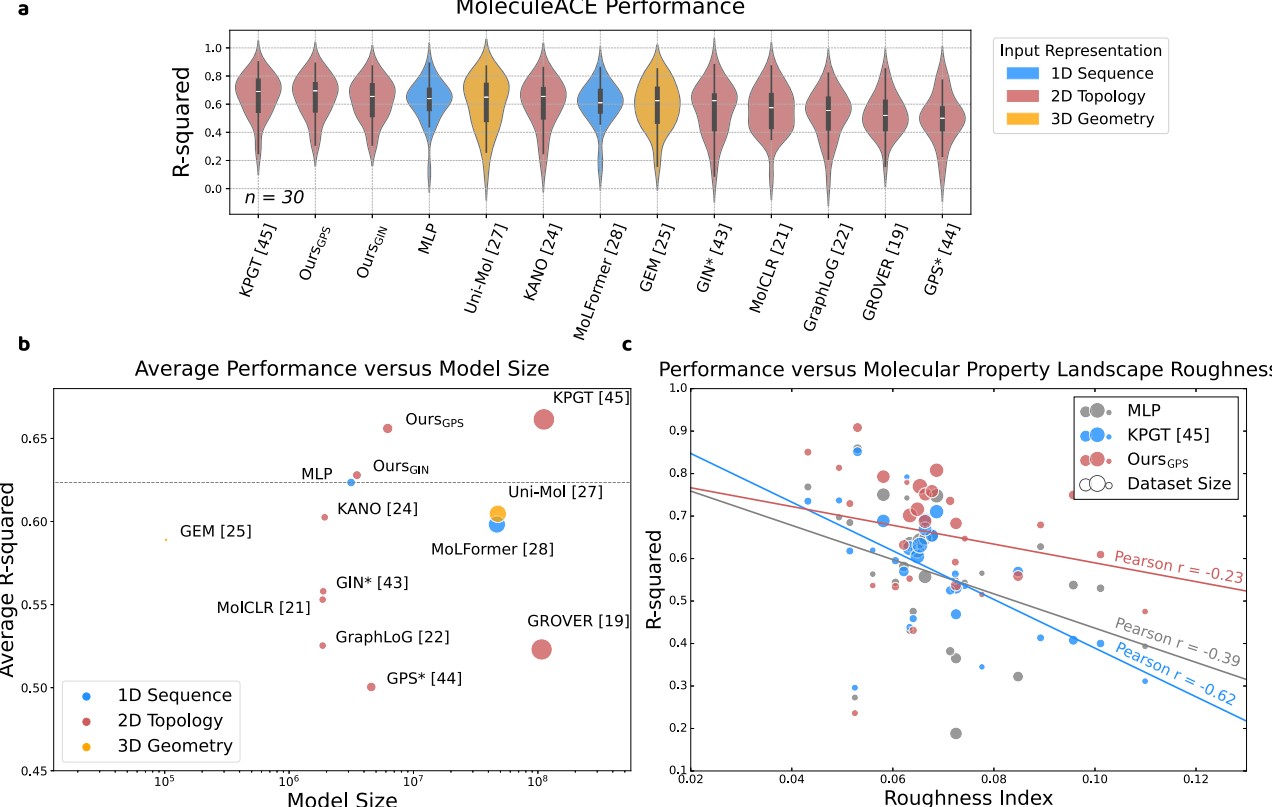

**Fig. 2 | Binding potency prediction. a** The violin plot illustrates model performance across 30 binding potency prediction tasks in MoleculeACE[6]. Performance on each dataset is averaged across three independent runs using stratified splits. The asterisk (*) indicates that the model is trained from scratch (no pre-train). The ordering of the methods from left to right is sorted by the average performance. The width of each violin represents the density of data points, with wider sections indicating a higher concentration of values. The minima and maxima are displayed by the lower and upper ends of the violin. The middle solid white line represents the median value. **b** Average model performance across 30 binding potency prediction tasks with respect to model sizes (also indicated by the size of the dots). The dashed line follows the performance of the multi-layer perceptron (MLP), which serves as the baseline comparison between traditional fingerprints and deep molecular representations. **c** Relationship between model performance and the presence of activity cliffs in each dataset measured by the roughness of molecular property landscape. Performance on each dataset is averaged across three independent runs using random splits. The size of the datasets is indicated by the size of the dots. The slope of the fitted line of each method indicates the correlation. Ours$_{GPS}$ is used in this analysis.

average distance between cliff MMPs to that of non-cliff MMPs. We also show three molecules on the plot for illustrative purposes, with one cliff MMP (indicated by the red arrow) and one non-cliff MMP (indicated by the blue arrow). 4. At last, we report the validation *R-squared* as the performance measure. Here are the three main takeaways from Fig. 3:

- Our composed representation yields the lowest ROGI value to start with. In other words, our pre-trained knowledge is more transferable to the target application. This is also shown by our rapid convergence rate in the validation set, reaching a validation R-squared of 0.676 at epoch 10 (Supplementary Fig. 5 and Supplementary Note 4).
- Our representation preserves the chemical knowledge learned from pre-training better than others, leading to less overfitting and better robustness. Since representations are optimized towards the property labels, the Rand index drops continuously for all methods. It means that the encoded information gradually shifts from being structure-oriented to label-oriented. However, our method has the lowest drop in Rand index of 0.072, compared to the drop of 0.09 by GraphLoG and 0.181 by MolCLR. The visualization also shows that the representations of MolCLR begin to overfit to the labels starting from epoch 10, resulting in the loss of substantial structural relationships between molecules. This explains its low ROGI value and validation R-squared along the training process.

- Our representation seems to exhibit a better understanding of the nuances of chemical knowledge in activity cliffs. Our average cliff-noncliff distance ratio in the validation set is always above one and larger than that of GraphLoG and MolCLR. As illustrated by the triplet samples, the red arrow is longer than the blue arrow across fine-tuning epochs, while the closeness of cliff and non-cliff pairs in space (i.e., structurally similar) is maintained. For MolCLR, even though the red arrow can be much longer than the blue one, the molecules are distant away. It means that MolCLR fails to capture the structural similarity aspect of the activity cliffs. A detailed examination of the distance histogram is visualized in Supplementary Fig. 13.

To further understand why our composed representation is more resilient to the representation shift and can better preserve pre-trained knowledge, we present more analysis on three diverse datasets, along with the individual channel-wise representation behavior during fine-tuning. We choose CHEMBL237_Ki and CHEMBL262_Ki as the representative regression-based datasets of different scales, and BBBP as the typical classification-based dataset. The Rand index of representation clustering difference between the initial and the current timestamp is computed as a proxy (slightly different than before) for the representation shift. A smaller Rand index indicates a larger shift. As illustrated in Fig. 4, channels with the

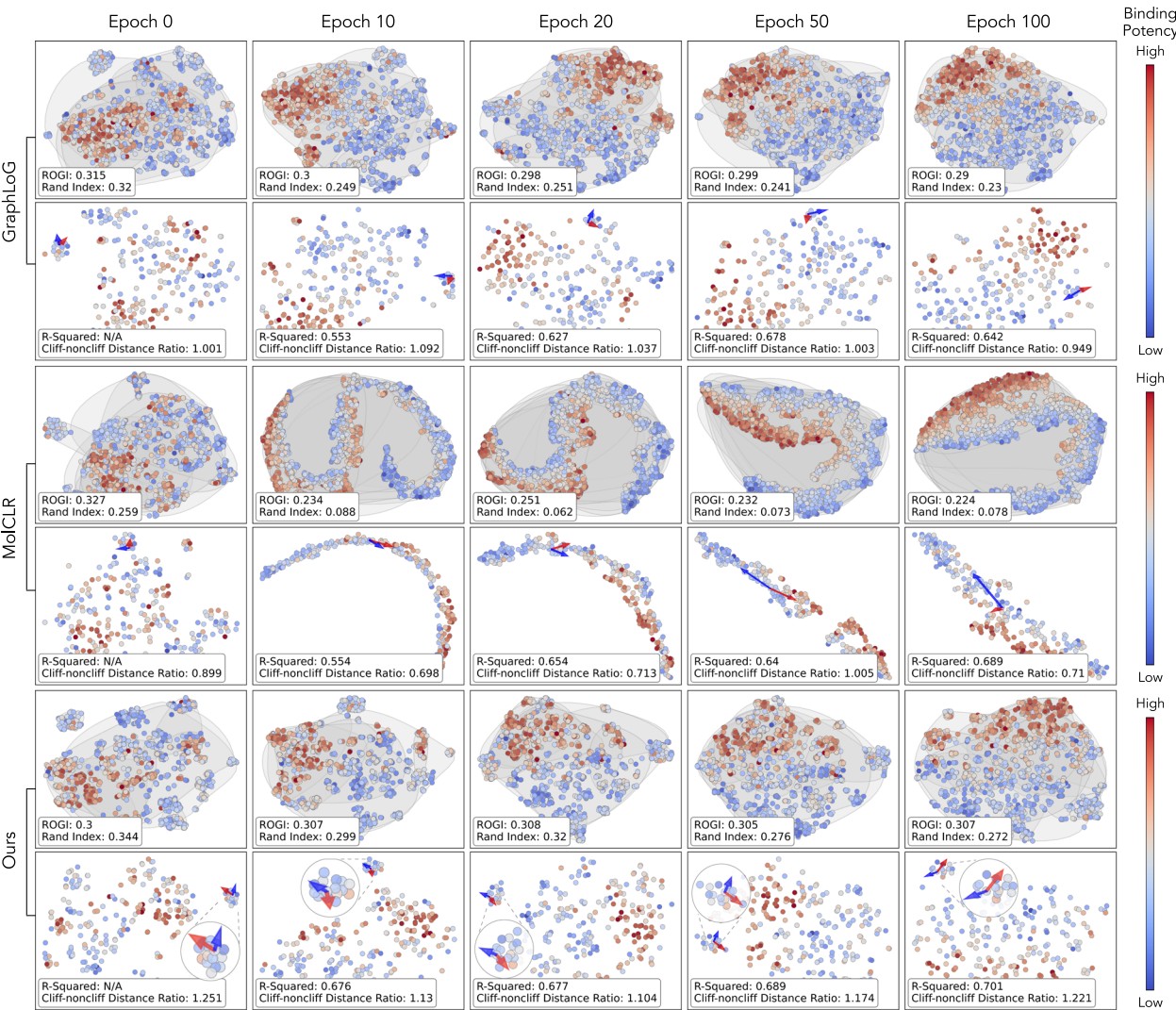

**Fig. 3 | Representation space probing.** The dynamics of molecule representation space of three methods at five finetune timestamps on dataset CHEMBL237_Ki ($n = 2602$). For each row pair, a 2D view of the representation space of both the training (top) and validation set (bottom) are visualized. The color map represents the normalized binding potency of each data point from lowest to highest within the dataset. Rand index and roughness index (ROGI) are reported for training, while R-squared value and cliff-noncliff distance ratio are reported for validation. The gray circles correspond to the clustering assignment using ECFP4 fingerprint. The red arrow indicates the distance of a cliff molecule pair, while the blue arrow indicates that of a non-cliff molecule pair.

highest prompt weights often exhibit the largest shift, and vice versa. This is reasonable because these channels contribute more to the optimization against the labels. Conversely, even though low-weighted channels contribute less to the optimization, they are more likely to preserve the pre-trained knowledge. As a result, the composed representation derived from channel aggregation exhibits a certain level of resilience to representation shifts, making it potentially more robust than other methods. The probing analysis under the few-shot learning settings also reveals similar patterns (Supplementary Fig. 6 and Supplementary Note 5).

**Activity cliffs analysis**

We present a deeper analysis into the potential of our method in understanding the activity cliffs. To be more specific, we evaluate the relationship between the model explanations generated by the GNNExplainer[49] and the predicted binding mode between the ligands and the protein pockets by AutoDock Vina[50]. Note that this analysis merely serves as a proof of concept, such that our representation has the potential of capturing influential and well-established factors in binding affinities. However, the fundamental limitation of utilizing topological information only is unavoidable, which we will discuss in Conclusion.

As shown in Fig. 5, visualized by PyMOL[51], two series of compounds sharing the same scaffolds are potential inhibitors of glycogen synthase kinase-3 beta (GSK3$\beta$). The molecule activity cliff pairs, determined by the formulation in[6], are compounds $< a1, a2 >$, $< a1, a3 >$, $< b1, b2 >$, and $< b1, b3 >$. The explanations of both our and MolCLR's predictions are compared. In Fig. 5a, the potential intra-molecular halogen-bonding contact between the chlorine atom and hydrazone in compound $a1$, as indicated by the green dashed line, is disfavored for the inter-molecular hydrogen-bonding contact between the backbone carbonyl of the active site VAL-135 and hydrazone of the compound[52]. As shown by our model explanation, compared to the compound $a2$ and $a3$, the chlorine atom of compound $a1$, along with its associated benzene ring, contribute less to the overall binding affinity prediction. It aligns well with the predicted binding mode.

The predicted binding mode in Fig. 5b shows that the orientation of the substituent (i.e., the alkoxy group in compound $b1$ at the 6 position of pyrazolo[1,5-b]pyridazine) will cause steric clash with PHE-67 in the G-rich loop of GSK3$\beta$, thereby leading to a loss of potency[53,54].

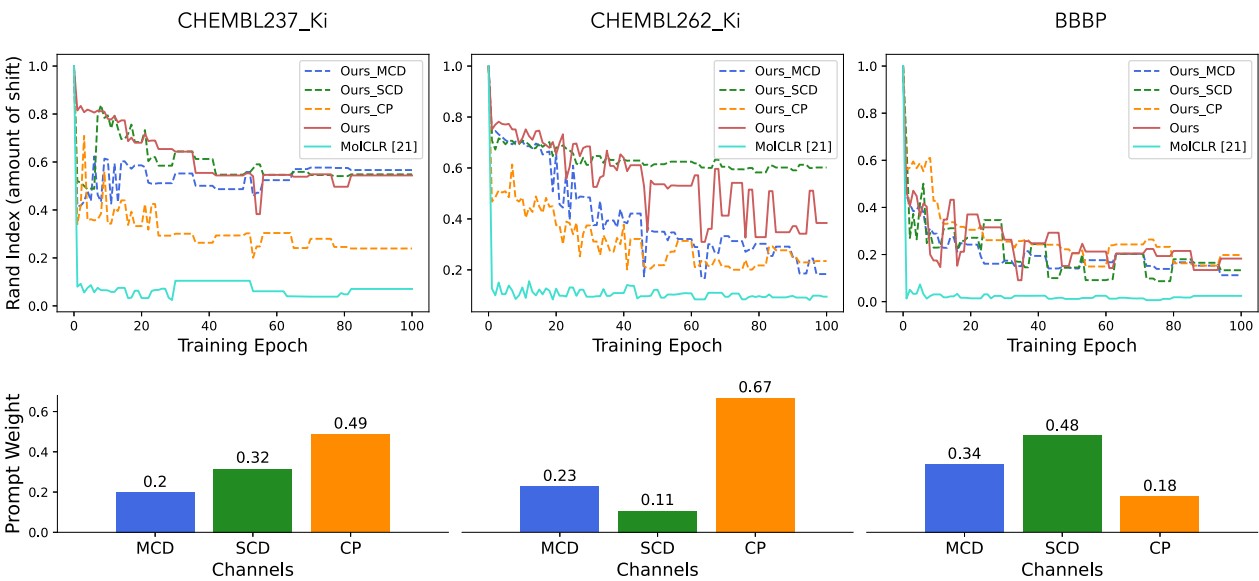

**Fig. 4 | Representation shift decomposition.** Detailed analysis of the representation shift during fine-tuning across three datasets. The dashed lines (top row) correspond to the representation shift (approximated by the Rand index) for each individual channel, while the solid lines represent the overall method. The bottom row displays the optimized prompt weights distributed over channels.

In contrast, the alkoxy groups at the 3' and 5' positions have no clear contact with the protein pocket. Remarkably, using only the topological information, our model can capture the importance of the influential alkoxy group at 6 position. In addition, the absence of the alkoxy group at 5' position (from compound *b*2 to *b*3) does not affect the overall atoms' contribution to the predicted potency. MolCLR performs equally good in terms of compound *b*2 and *b*3, but it fails to capture the influential alkoxy group in compound *b*1.

## Discussion

In this work, we propose a multi-channel learning framework for molecular representation learning, which aims to encode and utilize robust chemical knowledge that is generalizable to diverse downstream applications. Each channel is dedicated to learning unique self-supervised tasks, focusing on distinct yet correlated global and local aspects of the molecule. Specifically, MCD captures the molecule's global similarity, SCD highlights the fundamental role of the scaffold in affecting molecular characteristics, and CP targets the composition of functional groups. During fine-tuning, the model is able to identify which channel-wise representation is most relevant to the current application, thereby making the composite representation context-dependent.

One limitation of this framework is the need for a more effective prompt weight optimization mechanism. The initialization of prompt weights using roughness index can lead to sub-optimal performance. Since roughness index is a global QSPR metric that targets the overall chemical space, it does not account for any distribution shift between training and testing sets. This is the same for the other QSPR measures as well (e.g., SALI[55], SARI[56]). As a result, the final representation performance may be less correlated with the initial roughness value under designated splits. This also explains the performance gap between Fig. 2a, b.

There are several interesting directions for future research. One promising direction is to incorporate different input representations into the framework. By merely leveraging topological molecular structure, the model is unable to differentiate molecular components with different conformations (e.g., functional groups' orientation or atom's chirality), which could significantly alter biochemical behaviors

(see Supplementary Note 9 for further discussion). Besides, there exist other advanced data-driven techniques for studying the structural-activity relationship (SAR) that might be compatible with our framework. For example, Molecular Anatomy[57] argues that the network clustering from scaffold fragmentation and abstraction allows high quality SAR analysis. Such investigations aim to transfer knowledge from cheminformatics to machine learning models, potentially improving both model interpretability and robustness. More importantly, while our method has immediate implications for drug discovery, its molecular representation robustness further shed lights on its promising potential in other sub-fields of chemistry, such as materials science and environmental chemistry.

## Methods
### Graph neural networks (GNNs)

A graph $G = (V, E)$ is defined by a set of nodes $V$ and edges $E$. In the molecular graph, each node denotes an atom, and the edge denotes the chemical bond. Let $\mathbf{h}_v$ be the representation of node $v$, and $\mathbf{h}_G$ be the representation of the graph $G$. Modern GNNs follow the message-passing framework, such that node representations are updated iteratively via neighborhood aggregation:

$$\mathbf{h}_v^k = \text{UPDATE}\left(\mathbf{h}_v^{k-1}, \text{AGGREGATE}\left(\left\{\mathbf{h}_v^{k-1}, \mathbf{h}_u^{k-1}, e_{uv}\right\} : \forall u \in N(v)\right)\right),$$
(1)

where $N(v)$ is the neighborhood of node $v$, $k$ denotes the layer index in a multi-layer GNN structure, and $e_{uv}$ denotes the edge connecting two nodes $u$ and $v$. The initialization of $\mathbf{h}_v^0$ comes from the predefined node features $\mathbf{x}_v$. The aggregate function integrates neighborhood information into the current node representation. The update function takes the updated node representation and the node representation from the previous $k - 1$ layer and performs operations like concatenation or summation. After the iterative updates, a permutation-invariant pooling operation is performed to get the representation for the entire graph $G$: $\mathbf{h}_g = \text{READOUT}(\mathbf{h}_v^k | v \in V)$. There are various number of options for the readout operation, including simple operations of mean and max, and advanced differentiable approaches like DiffPool[58] and GMT[59].

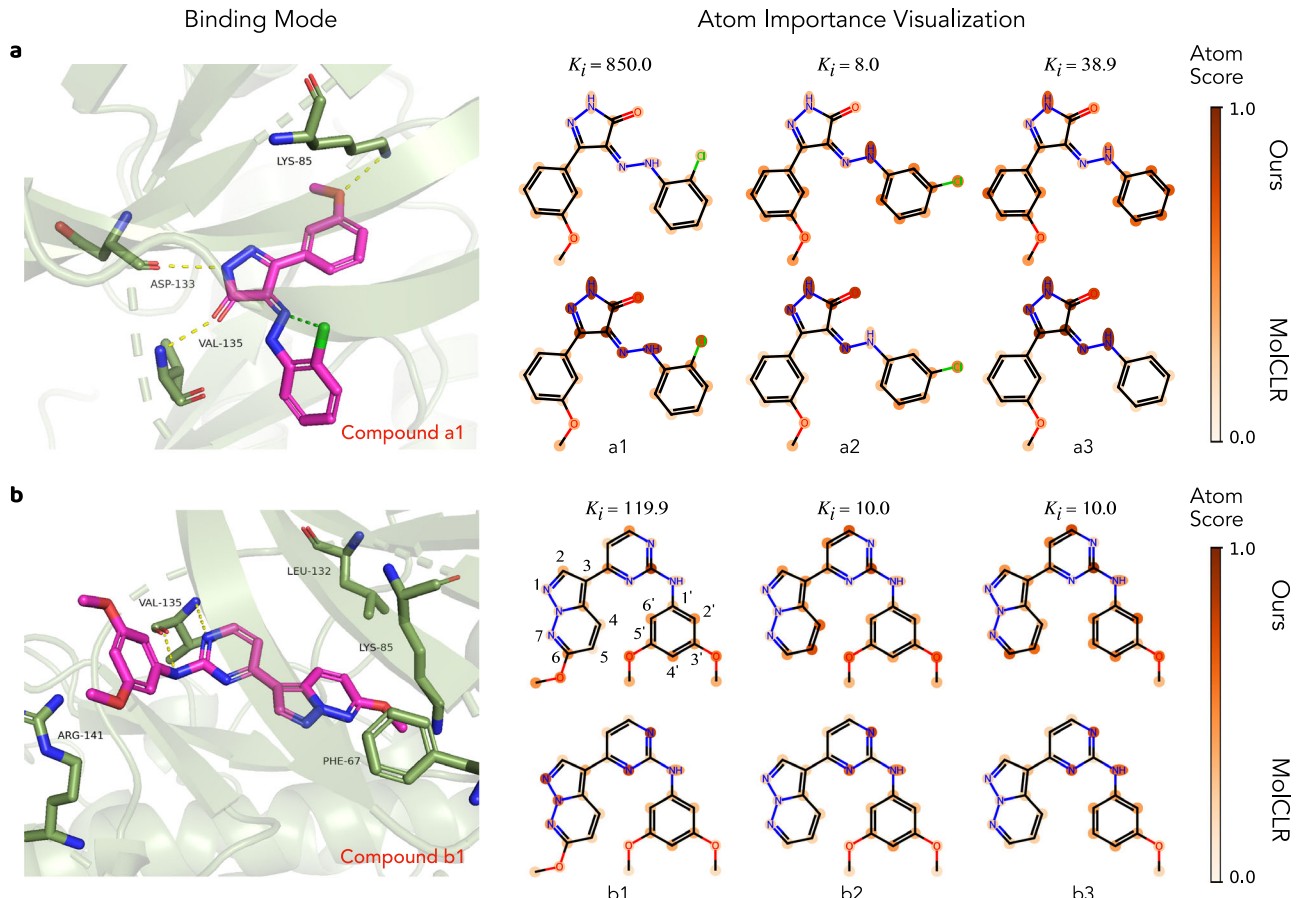

**Fig. 5 | Activity cliffs analysis.** Evaluate the bioactivity binding mode and the atom importance captured by our model on a (**a**) 5-phenyl-4-phenyldiazenyl-1,2-dihy-dropyrazol-3-one series and a (**b**) N-phenyl-4-pyrazolo[1,5-b]pyridazin-3-ylpyr-imidin-2-amine series. The binding modes of compound *a*1 and compound *b*1 in the active sites of GSK3*β*, with Protein Data Bank (PDB) ID: 3L1S, are predicted by AutoDock Vina[50] and visualized by PyMOL[51]. The key hydrogen bonds between compounds and the active sites are highlighted by yellow dash lines, while the green dash line refers to the intra-molecular halogen bond. The color intensity on each atom indicates its respective contribution to the prediction of our method, computed from a GNNExplainer[49].

## Roughness index

To analyze the structure-property relationship (SPR) within a chemical space, various quantitative metrics are proposed (e.g., SALI[55], SARI[56], ROGI[41]). Even though the formulations are different, they all aim to capture the relationship between the representation difference and the property difference, which is also known as the molecular property landscapes. In this work, we mainly rely on ROGI value for both model training and experimental analysis. It is computed by first clustering the chemical space with different distance threshold $t$. For each cluster assignment, the weighted standard deviation $\sigma_t$ is calculated over the property labels of each cluster prototype. As the distance threshold increases, $\sigma_t$ will decrease monotonically from its initial value to zero (i.e., single cluster). If there are more similar molecules with similar labels, $\sigma_t$ will decrease slowly, and vice versa. Eventually, roughness index is formulated as below:

$$\text{ROGI} = \int_0^1 2(\sigma_0 - \sigma_t)\, dt \qquad (2)$$

The ROGI value has been shown to be strongly correlated with other metrics (e.g., SARI), as well as the representation performance on the given dataset[41]. One of the main advantages of using ROGI is that it is a generalized QSPR metric applicable to a wide range of molecular representations, including both fingerprints and neural representations, while other metrics like SARI are tailored only for molecular fingerprints. This allows us to compute landscape roughness on pretrained molecular representations (Supplementary Fig. 4 and Supplementary Note 3) and perform deeper QSPR analysis.

## Prompt-guided aggregation

Instead of using the same readout operations as the conventional GNNs, we adopt the prompt-guided aggregation, which is achieved using the multi-head attention. As shown in Fig. 1e, the embedding of the prompt token $\mathbf{h}_p$ is treated as the attention query, while the representations $\mathbf{h}_x$ of nodes/atoms $x$, are viewed as the keys and values. The resulted prompt-aware graph representation $\mathbf{h}_g^p = \sum_x \alpha_x \mathbf{v}_x$, where the attention weight for each node $x$ is computed as $\alpha_x = \text{softmax}(\{\mathbf{q} \cdot \mathbf{k}_x / \sqrt{d_k}\}_x)$, and $\mathbf{q} = \mathbf{W}_q \mathbf{h}_p$, $\mathbf{k}_x = \mathbf{W}_k \mathbf{h}_x$, and $\mathbf{v}_x = \mathbf{W}_v \mathbf{h}_x$. Here $\{\mathbf{W}_q, \mathbf{W}_k, \mathbf{W}_v\}$ represent parameters in the linear transformations, and $\sqrt{d_k}$ is the scaling factor. Essentially, the prompt-aware $\mathbf{h}_g^p$ is aggregated from the weighted average of linear projection of $\mathbf{h}_x$.

## Adaptive margin contrastive loss

Contrastive learning is a technique widely used in self-supervised learning, which aims to group semantic similar samples closer while pushing dissimilar samples distant apart in the latent representation space. In this work, we adopt the triplet loss[40] to formulate the contrastive learning. It considers triplets of data samples: the anchor $G_i$, the positive (i.e., semantic-similar) sample $G_i'$, and the negative (i.e.,

semantic-dissimilar) sample $G_j$. Its formulation is shown below.

$$\ell_{i,j(\neq i)} = \max\left(0, \alpha + d\left(\mathbf{h}_{g_i}, \mathbf{h}'_{g_i}\right) - d(\mathbf{h}_{g_i}, \mathbf{h}_{g_j})\right), \qquad (3)$$

where $\mathbf{h}_{g_i}$ denotes the latent representation for sample $G_i$, and $\mathbf{h}'_{g_i}$ being the representation for its augmentation. The function $d(\cdot, \cdot)$ measures the L2 distance between two vectors. In general, this objective enforces the pair-wise distancing difference between $<G_i, G'_i>$ and $<G_i, G_j>$ to be at least $\alpha$ margin. However, as we discuss before, this formulation can only lead to a coarse-grained representation space, while neglecting the existing structural relationship among molecules (e.g., shared rings or functional groups). Also, this formulation does not pose any constraints on the representation space beyond the margin (Supplementary Fig. 3). Therefore, we propose to add an additional contrastive formulation with adaptive margin among negative samples and the anchor.

$$
\begin{aligned}
\ell_{i,j(\neq i),k(\neq j \neq i)} = &\max\left(0, \alpha_1(G_i, G_j) + d\left(\mathbf{h}_{g_i}, \mathbf{h}'_{g_i}\right) - d(\mathbf{h}_{g_i}, \mathbf{h}_{g_j})\right) \\
&+ \max\left(0, \alpha_1(G_i, G_k) + d\left(\mathbf{h}_{g_i}, \mathbf{h}'_{g_i}\right) - d\left(\mathbf{h}_{g_i}, \mathbf{h}_{g_k}\right)\right) \\
&+ \max\left(0, \alpha_2(G_i, G_j, G_k) + d\left(\mathbf{h}_{g_i}, \mathbf{h}_{g_j}\right) - d(\mathbf{h}_{g_i}, \mathbf{h}_{g_k})\right),
\end{aligned}
$$
$$(4)$$

where $\alpha_1(\cdot)$ and $\alpha_2(\cdot)$ are the adaptive margin functions. Let $\mathbf{z}_{g_i}$ be the conventional structural features of the sample $G_i$. We use the ECFP4 fingerprint[60], which hashes circular atom neighborhoods into fixed-length binary strings, to represent the structural features. In molecule contrastive distancing, the adaptive function $\alpha_1(G_i, G_j) = \alpha_{\text{offset}} \times (1 - \text{sim}(\mathbf{z}_{g_i}, \mathbf{z}_{g_j}))$, and $\alpha_2(G_i, G_j, G_k) = \alpha_{\text{offset}} \times (\text{sim}(\mathbf{z}_{g_i}, \mathbf{z}_{g_j}) - \text{sim}(\mathbf{z}_{g_i}, \mathbf{z}_{g_k}))$, where $\text{sim}(\cdot, \cdot)$ denotes the Tanimoto similarity. The scaffold contrastive distancing has the same formulation, except that the molecule sample $G$ is replaced by its scaffold $s(G)$. Note that the formulation now considers quadruplet of data samples $<G_i, G'_i, G_{j(\neq i)}, G_{k(\neq j \neq i)}>$. Even though the theoretical complexity is increased from $O(N^2K)$ to $O(N^3)$, we can perform fixed-size random sampling with respect to the computed similarity differences in $\alpha_2(\cdot)$. Quadruplets are also dropped if the computed values are negative. Eventually, the optimization goal is to minimize the loss summation as $\min \mathcal{L}_{adaptive} = \min \sum_{i,j,k} \ell_{i,j(\neq i),k(\neq j \neq i)}$.

## Regularization

To further improve the robustness of the pre-trained model, we incorporate two regularization schemes for the intra-channel node aggregation and the inter-channel alignment. The former strategy aims to ensure that all atoms contribute to MCD, while only the scaffold atoms contribute to SCD. This is accomplished by a smooth L1 loss between the attention score and the atom importance matrix. We realize that without any regularization, the aggregation module may rely on specific structural patterns to perform the SSL tasks, causing the attention distribution to skew towards certain substructures. This is also known as the shortcut learning[61] in deep learning models. As a result, it is possible to feed an incomplete view of the molecule to the property prediction layer (i.e., atoms that receive low attention across all channels) without the attention regularization. The latter regularization encourages better alignment of representation spaces. Since the three channels are learned separately via different tasks, the numeric values of representations at a given position may not be aligned. It means that the linear combination of representations before any fine-tuning may not be meaningful. To encourage the composite representation space to be better defined (e.g., subtracting $\mathbf{h}_g^{[MCD]}$ with $\mathbf{h}_g^{[SCD]}$ could represent the fragments beyond scaffold), we use a set of supervised tasks (e.g., predicting molecular weight and logP), along with the corresponding prompt weight presets, to regularize the

channel alignment. These tasks mainly predict the molecule/scaffold descriptors using the composite representation. For example, the prompt weight preset for predicting the molecular weight would be [0.45, 0.1, 0.45], while the preset for predicting the weight of the scaffold would be [0.1, 0.45, 0.45]. We multiply all the regularization losses by a factor of 0.1 to prevent the regularization from dominating the learned representation.

## Prompt-guided multi-channel learning

The overall multi-channel learning framework is inspired by the work in[62]. At the pre-train stage, the MCD and SCD channels perform the contrastive learning using the adaptive margin loss. Subgraph masking is used to generate positive samples for MCD. A subgraph is defined by a central atom along with its one-hop neighbors. The masking is performed at the attribute level, ensuring that the topological structure is retained. Meanwhile, the CP channel learns masked subgraph prediction as a multi-label classification task and motif prediction as a regression task: $\mathcal{L}^{\text{CP}} = \mathcal{L}_{\text{CE}} + \mathcal{L}_{\text{SmoothL1}}^{fg}$, where CE stands for cross-entropy loss, SmoothL1 for smooth L1 loss, and $fg$ for normalized functional group descriptors. Overall, the framework is optimized using the three channel losses along with the regularization losses:

$$\mathcal{L}_{overall} = \mathcal{L}_{adaptive}^{\text{MCD}} + \mathcal{L}_{adaptive}^{\text{SCD}} + \mathcal{L}^{\text{CP}} + 0.1 \times \mathcal{L}_{regu} \qquad (5)$$

At the finetune stage, the pre-trained model parameter are used to initialize the model. Besides, we introduce an additional prompt selection module to combine representations from different channels into the task-specific (i.e., context-dependent) composite representation. Essentially, it learns the relevance between different pre-trained molecular knowledge and the downstream application, hence bridging the gap between pre-training and fine-tuning objectives. To incorporate task-specific SPR information into the model at an early stage of fine-tuning, we propose to initialize the prompt weights from computing the roughness index (ROGI)[41]. In short, the initial prompt weights should lead to a composite representation with the lowest ROGI value (i.e., smoothest quantitative structure-property landscape) with respect to the current application. A low ROGI value indicates smoother landscape, hence better modellability. We use a simple Bayesian optimization pipeline to find the best initialization with the lowest ROGI value on the training set. Essentially, the input parameters of the Bayesian optimization are $k - 1$ learnable scalers, where $k$ is the number of channels. The black-box utility function first computes the distribution over $k$ values by treating the input scalars as logits, and calculates the composite representation as well as its corresponding roughness index. We utilize the Quasi MC-based batch Expected Improvement as the acquisition function. After the initialization, the entire model except for the prompt-guided aggregation module is optimized towards the molecular property labels. The parameters in the aggregation modules are fixed during fine-tuning as we aim to use it as a pooling layer independent of the downstream applications. As shown in Supplementary Figs. 7–9, the node attention scores within the aggregation module reflect the expected atom importance. For example, the SCD's aggregation attention captures mostly the scaffold atoms. We further discover that the learned prompt weights exhibit patterns that are well-aligned with the necessary chemical knowledge for solving the downstream tasks (Supplementary Fig. 12, Supplementary Table 1, and Supplementary Note 8).

## Experimental setup

We pre-train our framework using the molecules from ZINC15[38], which is an open-sourced database that contains 2 million unlabeled drug-like compounds. We use CReM[63], an open-sourced molecule mutation framework, for the scaffold-invariant molecule perturbation. To be more specific, we first use RDKit[64] to identify the Bemis-Murcko scaffold of molecules, as well as the connection sites (i.e., atom indices)

between the scaffold and the side chains. The CReM algorithm then takes these indices and performs fragment replacement using an external fragment pool. Perturbation results are further filtered by chemical validity and the maximum number of changed atoms allowed. At last, we are able to successfully compute perturbations for 1,882,537 out of 2 million molecules. These molecules form our final pre-train database. In the contrastive-based pre-training, we consider five positive samples for each anchor molecule. In molecule distancing, we randomly apply subgraph masking five times on the original molecule. In scaffold distancing, we randomly sample five scaffold-invariant perturbations. In terms of the model architecture, we follow the same architecture setup as the graph neural network in[18,21–23], which is a 5-layer Graph Isomorphism Network (GIN)[42] with hidden dimension size equals to 300. In addition, we also pre-train a GPS Graph Transformer model[43] with the same number of layer and hidden dimension size. We use the same molecule feature sets as works in[19,24]. During fine-tuning, we utilize the scaffold splits method[18] for MoleculeNet[5], with a train, validation, and test ratio of 8:1:1. On the other hand, we apply the stratified splits for MoleculeACE[6], as proposed by itself, with a train, validation, and test ratio of 8:1:1.

## Baselines

We consider twelve baseline pre-training methods in our experiments. 1. GraphLoG[22] achieves a hierarchical prototypical embedding space with the conventional graph perturbation techniques; 2. D-SLA[23] proposes the discrepancy learning to refine the embedding space with the conventional graph perturbation techniques; 3. MolCLR[21] utilizes the NT-Xent[33] contrastive loss with molecule perturbation techniques, including atom/bond editing and subgraph masking; 4. KANO[24] introduces the knowledge graph prompting techniques to augment molecular graph, while the augmentations are also learned in the contrastive manner. 5. MolFormer[28] performs large scale of masked language modeling on SMILES sequence using a linear attention Transformer. 6. Hu et al.[18] adopts masked attribute prediction and molecular subgraph prediction; 7. KPGT[44] proposes the Line Graph Transformer on molecular graphs and performs knowledge prediction. It takes a masked molecular graph, molecule fingerprint, and molecule descriptors as input, and aims to reconstruct the molecule fingerprint and descriptors during pre-training. 8. GROVER[19] proposes the GTransformer architecture and applies context prediction learning on molecular graphs; 9. GEM[25] encodes the 3D geometric information of molecules and predicts the bond angle and atom distance. 10. Uni-Mol[27] also incorporates 3D information of the molecule and pre-trains a SE(3) Transformer[65] using 3D position recovery and masked atom prediction. We further include 11. GraphMVP[45], which performs both contrastive and predictive learning between 2D toplogical and 3D geometric information of the molecule. 12. ImageMol[46], which leverages visual information of molecule images for molecular property prediction. We also train the GIN[42] and GPS[43] models from scratch for comparison. We reproduce the performance of MolCLR, GROVER, GEM, KPGT, MolFormer, and KANO on the MoleculeNet benchmark using their provided checkpoints and code repositories. Note that the reported results of GROVER, KPGT, MolFormer and KANO in their original papers are evaluated under the balanced scaffold splits, which is different than the deterministic scaffold splits used in this work and the rest of the baseline methods. Both splits aim to hold out a set of molecules with scaffolds that are not seen during training. The deterministic scaffold splits prioritizes the satisfaction of the specified split ratio, while the balanced scaffold splits prioritizes the balance of scaffold frequency across the training, validation, and test sets. We use the deterministic scaffold splits in our experiments primarily because most of the baselines also employ it. All other aspects, including hyperparameter choices for both the models and the training setup, remain consistent with the default configurations of these baseline methods for replicating the results. For the MoleculeACE benchmark, we leverage their corresponding repositories, default

configurations, and the provided model checkpoints of these baseline methods and fine-tune each task using either MoleculeACE's stratified splits or random splits.

## Representation space probing

We propose to analyze the dynamics of representation space during fine-tuning to evaluate the representation robustness and generalizability. To be more specific, we probe the mapping of the representation space at five different training timestamps (epoch 0, epoch 10, epoch 20, epoch 50, epoch 100). The 2D view is constructed via the T-SNE[66] dimension reduction technique. Besides the visualization, we also report four additional metrics that capture the representation characteristics. We report the ROGI value and Rand index in the training set, plus the R-squared value and cliff-noncliff distance ratio in the validation set. The Rand index is calculated from two clustering assignments. The first clustering is formed using k-means clustering with the molecule ECFP4 fingerprint (radius=2, nBits=512). The second clustering is formed using k-means clustering with the representation at the current timestamp. We set the same number of clusters for these two clusterings. In terms of the cliff-noncliff distance ratio, we first identify the cliff and the noncliff molecule pairs using the same principle in MoleculeACE[6]. Then, we compute and average the representation distance of each pair. At last, the cliff-noncliff ratio is formed by normalizing their average distance.

## Reporting summary

Further information on research design is available in the Nature Portfolio Reporting Summary linked to this article.

## Data availability

All relevant data supporting the key findings of this study are publicly available. All datasets utilized in this paper can be found at https://github.com/yuewan2/MolMCL or the Zenodo repository at https://doi.org/10.5281/zenodo.14010436[67]. This includes the processed pre-train dataset of ZINC15, the MoleculeNet benchmark (originally from https://github.com/deepchem/deepchem) and the MoleculeACE benchmark (originally from https://github.com/molML/MoleculeACE). The reference protein structure for 3L1S used in this study is available in the Protein Data Bank under accession code 3L1S. Source data are provided with this paper.

## Code availability

The source code of this work can be accessed via the GitHub repository at https://github.com/yuewan2/MolMCL or the Zenodo repository at https://doi.org/10.5281/zenodo.14010436[67].

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

## Acknowledgements

We have no specific acknowledgments to report for this work.

## Author contributions

Y.W. proposed and implemented the methodology, and conducted the computational experiments. J.W. carried out the chemical analysis of the model's predictions. T.H., C.H., and X.J. provided guidance throughout the project. Y.W. and J.W. led the paper writing and all authors contributed.

## Competing interests

The authors declare no competing interests.
