## [Peer Review File · Nature Communications]

REVIEWER COMMENTS

Reviewer #2 (Remarks to the Author):

Accurate prediction of molecular properties is paramount in the field of drug discovery. Deep learning-based methods, particularly those leveraging self-supervised learning, have gained widespread adoption for enhancing molecular representation learning and thereby facilitating accurate prediction of various molecular properties. In the manuscript "From molecules to scaffolds to functional groups: building context-dependent molecular representation via multi-channel learning", Yue et al, introduce a self-supervised learning framework for molecular representation learning. This framework incorporates three learning objectives: molecule contrastive distancing, scaffold contrastive distancing, and context prediction, with the aim of bridging the gap between domain-agnostic self-supervised learning (SSL) methods and domain-specific molecular tasks.

While the manuscript is well written, the novelty of this work appears somewhat limited. The components of this learning framework seem to be primarily derived from previous works, and the pipeline itself does not present significant novelty. Furthermore, the manuscript does not convincingly address the core challenges it discusses, and the effectiveness of the proposed method in predicting molecular properties was not thoroughly evaluated. The following are specific comments and suggestions for improvement.

Major comments:

1. The major components of the proposed framework are adopted from previous studies, such as the objectives (molecule contrastive learning, motif prediction, and masked subgraph prediction) and the prompt-based aggregation module. Moreover, the prediction performance of the proposed framework is also less than ideal, as it achieved comparable or worse performance on 4 out of 7 datasets from MoleculeNet benchmark. Therefore, this work seems to provide limited contributions to this field.
2. As discussed in the Introduction section, molecules might lose their characteristics by molecule perturbation. The learning objectives employed in the proposed method, such as molecule contrastive distancing, scaffold contrastive distancing, and context prediction, all stem from this molecule perturbation concept. Therefore, a contradiction exists between the motivation and methodology. Moreover, I think that this discrepancy is the cause of the observed lack of significant improvement over baseline methods in the proposed approach.
3. The prediction performance of the proposed method need to be further evaluated. The authors discussed that current self-supervised learning-based methods on molecules perform worse than the molecule fingerprint-based methods in the Introduction section. Therefore, molecule fingerprint-based methods should be included as baseline methods in the computational tests. Moreover, key self-supervised learning baseline methods are missed in the benchmarking tests, including KPGT[1], ImageMol[2], and MolFormer[3]. More baseline methods should also be incorporated into the benchmarking tests on the datasets from MoleculeACE.

4. The authors claim that the proposed method prioritizes structure over labels during fine-tuning (line 255, page 8). However, Figure S5 reveals a consistent tendency for the proposed method to rapidly overfit the training set, contradicting the claimed structure-oriented feature. It is unclear whether a multiple-layer perceptron (MLP) is utilized for predictions on top of the pre-trained model. If so, the comparison in Figure 2 may not be equitable. Despite the pre-trained model generating structure-oriented molecular representations (due to the fixed prompt-guided aggregation module during training), the inner layer of the MLP could introduce label-oriented aspects.

5. Given the incorporation of numerous components in the proposed framework, it is essential to conduct comprehensive ablation studies to assess the necessity of each element.

6. It is imperative to make the codes and datasets employed in this study publicly available. This accessibility is crucial not only for evaluating the model but also for the broader utility within the scientific community.

Minor comments:

1. There are typos in the manuscript

In line 252, "can be better transferred to the target application" should be revised to "can be better transferred to the target application"

In line 265, "the red arrow is always larger than the blue arrow" should be revised to "the red arrow is always longer than the blue arrow".

2. Additionally, is it correct to state "Our cliff-noncliff distance ratio in the validation set is always positive"? As per my understanding, the cliff-noncliff distance ratio should indeed always be positive.

3. The arrows in Figure 2 do not effectively convey their intended meaning. Consider using alternative visual representations for clarity.

4. The definitions of "bonded results" and the term "+-" should be explicitly provided in the captions of Tables 1 and 2 to enhance clarity.

References

[1] Li H, Zhang R, Min Y, et al. A knowledge-guided pre-training framework for improving molecular representation learning[J]. *Nature Communications*, 2023, 14(1): 7568.

[2] Zeng X, Xiang H, Yu L, et al. Accurate prediction of molecular properties and drug targets using a self-supervised image representation learning framework[J]. *Nature Machine Intelligence*, 2022, 4(11): 1004-1016.

[3] Ross J, Belgodere B, Chenthamarakshan V, et al. Large-scale chemical language representations capture molecular structure and properties[J]. *Nature Machine Intelligence*, 2022, 4(12): 1256-1264.

Reviewer #3 (Remarks to the Author):

The authors introduce a novel learning framework that leverages the knowledge of structural hierarchies within molecular structures embeds them through separate pre-training tasks over distinct channels, and employs a task-specific channel selection to compose a context-dependent representation. The work aims to enhance molecular representation learning while bridging the gap between domain-agnostic self-supervised methods and domain-specific molecular tasks. This approach demonstrates competitive performance across various molecular property benchmarks and establishes some state-of-the-art results. The authors claim that this method has unprecedented advantages in particularly challenging yet ubiquitous scenarios like activity cliffs with enhanced robustness and generalizability compared to other baselines. After significant revisions, the manuscript would potentially emerge as a significant contribution to the field of Cheminformatics, rendering it a potential candidate for consideration in Nature Communications. Detailed critiques and suggestions concerning the manuscript are delineated subsequently.

1. I have carefully reviewed your manuscript and the associated code repository. It is commendable to observe that the repository is well-organized and adequately structured, which significantly contributes to the transparency and reproducibility of your research findings. The clear organization and documentation facilitate easy understanding and use of your data and code, which is essential for advancing research in your field. However, I noticed an omission that, if addressed, could greatly enhance the utility and accessibility of your repository. Specifically, the repository lacks a detailed description of the dependencies (requirements.yml) required to run your code across different operating systems, namely iOS, Linux, and Windows. This oversight could pose significant barriers to reproducibility and collaboration, especially for researchers working in diverse computing environments.

2. Line 95-96: "The overall framework is pre-trained using ZINC15 [33], and evaluated on 7 molecular property prediction tasks in MoleculeNet [46] and 30 binding potency prediction tasks in MoleculeACE". It's important to clarify that MoleculeNet comprises benchmarks rather than tasks. For instance, benchmarks like Tox21 and ToxCast encompass multiple tasks within them, not just a single task. This distinction is crucial for accurately describing the evaluation methodology and the scope of the framework's applicability.

3. Line 124: Please define the acronym MolCLR: Molecular Contrastive Learning of Representations).

4. Lines 148-151: "In addition, a prompt-tuning module $\tau\theta(\cdot)$ is introduced in fine-tuning to determine which channel is most relevant to the current application. It essentially learns a task-specific prompt distribution. The prompt weights are utilized to linearly combine the channel-wise information into a composite molecule representation, which is then used for the task-specific prediction." This point caught my attention. I particularly think it is important to incorporate analyses of toxicological properties (e.g., AMES) with known structural alerts and Modes of Action into the manuscript. This way, you could explain how the method performs in these scenarios.

5. Line 151: "We initialize the prompt prompts by the choosing the candidate". Correct the sentence: "We initialize the prompt by choosing".

6. Line 252: "our pre-trained knowledge can be better transferedto the target". Correct the sentence:

“our pre-trained knowledge can be better transferred to the target”.

7. General Comment: The criteria for classifying a pair of molecules as an activity cliff need clarification. Additionally, examining the chemical spaces of the datasets under investigation seems crucial to ascertain the presence of activity cliffs, specifically their global SAR discontinuity. I suggest conducting this analysis using methodologies like SARANEA (<https://pubs.acs.org/doi/10.1021/ci900416a>), which enables the determination of the chemical space's level of discontinuity through a SAR Index (SARI). The global SARI is a numerical function designed to assign a SAR category to given sets of active compounds. Possible values range from 0 to 1, and low, intermediate, and high values reflect three general SAR types, discontinuous, heterogeneous, and continuous SARs, respectively.

8. Lines 284-286: "To be more specific, we evaluate the relationship between the model explanations generated by the GNNExplainer [51] and the predicted binding mode between the ligands and the protein pockets by AutoDock Vina". Given the framework's demonstrated potential in evaluating molecular properties and binding potency predictions, it is crucial to incorporate interpretable explanations code into the framework's repository. While it is commendable that the authors have utilized GNNExplainer for interpretation, the omission of this from the repository limits the framework's transparency and utility. GNNExplainer, being a model-agnostic tool, offers interpretable explanations for predictions made by any Graph Neural Network (GNN)-based model on graph-based machine learning tasks. By integrating GNNExplainer or similar tools directly into the repository, you significantly enhance the framework's reproducibility and enable users to gain deeper insights into the predictive mechanisms. This inclusion not only bolsters the scientific rigor of your work but also facilitates broader application and understanding of your model's predictions, reinforcing the value of your contributions to the field.

9. Line 361: "We use the Morgan fingerprint [25], which hashes circular atom neighborhoods into fixed-length 352 binary strings, to represent the structural features". It is crucial to use the nomenclature ECFP4 instead of "Morgan fingerprints" in your manuscript for two reasons. Firstly, ECFP4, or Extended Connectivity Fingerprints with a diameter of 4, is a specific type of circular fingerprint widely recognized in cheminformatics for its effectiveness in encoding molecular structure information. The term "ECFP4" precisely describes the algorithm and parameters used, facilitating clear communication and understanding among researchers in the field. Secondly, while "Morgan fingerprints" refer to a broader class of circular fingerprints developed by the Morgan algorithm, the term does not specify the diameter or other critical parameters that define the fingerprint's resolution and scope. Using the more specific "ECFP4" ensures that readers and fellow researchers can accurately reproduce your work, understand the basis of your computational analyses, and compare your results with other studies using the same or similar molecular descriptors.

10. Lines 59-61: "Meanwhile, several studies [40, 9, 45] have demonstrated that existing pre-trained molecular representations struggle with challenging yet ubiquitous situations like activity cliffs in drug discovery". Although Nature Communications is a journal with a significant impact factor, it is not specialized in Cheminformatics or Medicinal Chemistry. Explain the concept of "Activity Cliffs" and their impacts on predictive modeling.

11. Page 9: Regarding Figure 4, could you clarify the molar mass units used for the depicted compounds?

Are they in nanomolar (nM) or micromolar (μM)? Additionally, I noticed that the potency values are normalized between 0 and 1, a method that seems unusual in the QSAR domain. Typically, potency values (K_i and IC_{50}) are normalized using a negative logarithmic scale to calculate $\text{p}K_i$ and pIC_{50} values. Could you explain the reasoning behind this unconventional normalization technique?

12. In your manuscript's methodology section, I'm curious about the decision not to consider or utilize alternative self-supervised learning architectures, particularly BERTs (Bidirectional Encoder Representations from Transformers). Given the notable advancements and proven effectiveness of transformer-based models like MolBERT in cheminformatics for predicting molecular properties, understanding the rationale behind your chosen computational approach would be valuable.

13. Table 2: Some SSL methods from Table 1 were not included in Table 3. Is there any justification for this?

14. General comment: In the Methods section, please ensure proper citation of the methods introduced in the introduction, including MolCLR, GROVER, GEM, and KANO. Additionally, it's essential to detail the Roughness Index equation within this section, as it represents a less common metric.

15. Lines 411-412: The term "KMeans" is commonly used in software implementations and programming libraries to refer to the K-means clustering algorithm. However, in academic and theoretical discussions, the lowercase "k-means" is often preferred to maintain consistency with mathematical notation and the general convention of describing algorithms.

16. Support information, line 16: In your manuscript, you refer to "tanimoto similarity." Given the convention in scientific literature, the term should be capitalized as "Tanimoto similarity" to properly acknowledge Tanimoto's contribution to the field.

17. Lines 404-416, Representation space probing section: Could the authors please make the script for the "Representation Space Probing" analysis available in a public repository? This would greatly facilitate the understanding and reproducibility of your methodology, allowing for a more in-depth evaluation of your approach.

Reviewer #3 (Remarks on code availability):

I have carefully reviewed your manuscript and the associated code repository. It is commendable to observe that the repository is well-organized and adequately structured, which significantly contributes to the transparency and reproducibility of your research findings. The clear organization and documentation facilitate easy understanding and use of your data and code, which is essential for advancing research in your field. However, I noticed an omission that, if addressed, could greatly enhance the utility and accessibility of your repository. Specifically, the repository lacks a detailed description of the dependencies required to run your code across different operating systems, namely iOS, Linux, and Windows. This oversight could pose significant barriers to reproducibility and collaboration, especially for researchers working in diverse computing environments.

Reviewer #4 (Remarks to the Author):

The paper presents a novel molecular representation learning framework aimed at enhancing the reliability and adaptability of molecular property prediction, which is of great significance to broad research fields such as chemical science and drug discovery. It achieves context-dependent molecular representations by multi-channel learning within the chemical landscape hierarchy including functional groups, scaffolds, and global characteristics. Experimental results highlight the representation robustness and the ability to handle activity cliffs.

However, several issues should be addressed before the manuscript is ready for publication to support the conclusions. See below:

- (1) The experiment results on molecular property prediction and binding potency prediction lacks thorough analysis. The authors should provide insights about the superior performance of the model, report statistical significance test results, and discuss the limitations of the model.
- (2) The results on ToxCast are missing for MoleculeNet.
- (3) The authors should compare their model with more baseline approaches such as GraphCL[1], GraphMVP[2], and UniMol[3] on both tasks.
- (4) The authors should provide more analysis on the context-dependent molecular representations and the optimized prompt weights by answering the following questions:
 - (4.1) Does the attention maps of the aggregation modules reflect the key features of functional groups, scaffolds, and the global structure?
 - (4.2) How does the representations of each channel affect downstream performance?
 - (4.3) What's the patterns of the prompt selection module on different fine-tuning datasets? Do they align with the chemical knowledge required to solve corresponding tasks?
 - (4.4) How does the prompt weight initialization trick aid the local optimum problem?
- (5) For activity cliff analysis, the authors claimed in L265-266 that "the red arrow is always larger than the blue arrow". However, the comparison seems insignificant in Figure 2. The distribution of cliff-noncliff distance ratio should be visualized.
- (6) The work is focused on 2D molecular graphs of molecules, which is credible. However, the authors should provide a thorough analysis of different structural representations of molecules including 1D SMILES strings, 2D molecular graphs and 3D conformations. Consider answering the following questions:
 - (6.1) What's the performance comparison between the proposed model and those that incorporate 1D or 3D information?
 - (6.2) Is the proposed method applicable to 1D and 3D molecular data?

The novelty and soundness of the proposed methodology is fair. However, I do have a question about the fine-tuning process:

- (7) In L378-379, the authors claimed that "the entire model except for the prompt-guided aggregation modules is optimized". However, in Figure3 it seems that the GNN representations exhibit shifts to a notable extent. The authors should investigate if the aggregation patterns still hold for task-oriented representations.

The manuscript is overall clearly written. However, several parts of the manuscript should be clarified:

(8) In L107, the authors claimed the visualization of chemical space clustering, which is presented in Supplementary Information. Consider moving these results in the main manuscript.

(9) The definition of "cliff-noncliff distance ratio" at L244-L247 is not clear.

(10) The authors should provide more information about the pre-training objectives, including mathematical forms and calculation of the overall objective.

(11) L252: transferto -> transfer to

Finally, we encourage the authors to improve the reproducibility of the manuscript:

(12) Provide more information about the implementation details, including hyper-parameters of the pre-training and fine-tuning experiments, as well as computational costs.

(13) Open-source the data and code to make the work easier to follow.

To all reviewers:

We would like to thank all reviewers for their time and efforts in reviewing our manuscripts, as well as the constructive feedback and valuable suggestions. We believe most concerns have been addressed adequately.

Before we proceed to the point-to-point response, we would like to first make a general overview of the main adjustments we have made compared to the original version, which is listed below:

1. In addition to applying the proposed prompt-guided learning framework to GNN, we also pre-train a Graph Transformer model using the proposed framework to test its effectiveness across different model architectures.
2. We incorporate two regularization tricks into the pre-training (discussed in the Method section).
3. As the reviewer suggested, we added more baseline methods for both the MoleculeNet and MoleculeACE benchmarks. These baseline methods are diverse in terms of pre-training strategies, input representations (e.g., SMILES/Graph/3D geometry), model backbones (GNN vs Transformer-based), and model sizes.
4. We performed more qualitative analysis in terms of the prompt-guided node aggregation and the prompt-guided channel selection (included in Supplementary Information).
5. We provided more discussion on the model performance comparison and included more implementation details.
6. We also resolved the writing issues mentioned by the reviewers.

The code repository is now published at <https://github.com/yuewan2/MolMCL/tree/main>, which includes the detailed instruction and our hyperparameter choices for both pre-training, fine-tuning, visualization of prompt aggregation patterns, and the representation probe. We are also working rapidly to clean up the code for the rest of the qualitative analysis.

To reviewer #2:

We thank you for your valuable time and efforts in reviewing our manuscripts, and we hope that your concerns could be addressed by the following response.

1. Response to the framework novelty and the limited performance in MoleculeNet

The main goal of this work is to build a generic and highly generalizable molecular representation that is not only suitable to learn molecular properties across a diverse range of molecular scaffolds, but also captures the subtle variations across fine-scaled

structural distinctions (such as the phenomenon of activity cliffs) among highly similar molecules. One major innovation of the proposed method is its flexibility to combine molecule representations at different levels (overall structure, scaffold, functional group) according to the needs of downstream prediction tasks. The proposed method contrasts with existing methods that focus on a single type of representation, which improves its adaptability to various prediction tasks.

Additionally, as we discuss the issues with existing methods in the Introduction section, we further refine the learning method for each type of molecule representation based on the chemical knowledge. We leverage contrastive learning in our framework since it is well aligned with the common heuristics in chemistry and with our goals. However, our contrastive formulation, built on top of the adaptive triplet loss, is largely different from the previous contrastive methods that utilized the NT-Xent loss (e.g., MolCLR, KANO). We also propose a novel scaffold-invariant molecular perturbation method, which no one has done before. Also, the conventional motif prediction is framed as the multi-label binary classification task, which is shown to be less helpful than other SSL tasks [1]. Therefore, we adjust it into a regression task that predicts the normalized functional group descriptors and improves model learning.

Finally, an important contribution of this work is the development of an effective model for molecular property prediction that naturally handles the issues of activity cliffs. We argue that it is essential to consider the performance from both benchmarks (i.e. MoleculeNet and MoleculeACE) simultaneously to evaluate the quality of the pre-trained representations. Even though our method does not significantly improve the SOTA performance in MoleculeNet, its average performance, 80.6 in AUC, is **still the highest** among the other strong baselines. The second best method is Uni-Mol, which has an average AUC score of 79.8, but is an order of magnitude larger than our model. Moreover, Uni-Mol cannot competitively handle issues of activity cliffs as illustrated in a

later figure below.

Table 1: Fine-tuning results on 7 classification tasks in MoleculeNet. Average ROC-AUC value is reported, along with the score standard deviation from three independent runs. The first two rows (GIN and GPS) show the results under the no-pretrain setting.

Methods	BBBP	Clintox	MUV	HIV	BACE	Tox21	SIDER	Avg.
#task	1	2	17	1	1	12	27	
GIN [57]	65.8 ± 4.5	58.0 ± 4.4	71.8 ± 2.5	75.3 ± 1.9	70.1 ± 5.4	74.0 ± 0.8	57.3 ± 1.6	67.5
GPS [35]	64.8 ± 3.0	87.2 ± 0.9	69.8 ± 3.8	73.1 ± 3.3	78.0 ± 3.0	74.5 ± 0.6	60.8 ± 0.6	72.6
Hu et. al [21]	70.8 ± 1.5	72.6 ± 1.5	81.3 ± 2.1	79.9 ± 0.7	84.5 ± 0.7	78.7 ± 0.4	62.7 ± 0.8	75.8
GraphLoG [58]	72.3 ± 0.9	74.7 ± 2.2	74.2 ± 1.8	75.4 ± 0.6	82.2 ± 0.9	75.1 ± 0.7	61.2 ± 1.1	73.6
D-SLA [24]	72.6 ± 0.8	80.2 ± 1.5	76.6 ± 0.9	78.6 ± 0.4	83.8 ± 1.0	76.8 ± 0.5	60.2 ± 1.1	75.5
MolCLR [50]	73.5 ± 0.4	90.4 ± 1.7	75.5 ± 1.8	77.6 ± 3.2	83.5 ± 1.8	76.7 ± 2.1	60.7 ± 5.7	76.8
GraphMVP [27]	72.4 ± 1.6	79.1 ± 2.8	77.7 ± 0.6	77.0 ± 1.2	81.2 ± 0.9	75.9 ± 0.5	63.9 ± 1.2	75.3
GROVER [37]	69.5 ± 0.1	76.2 ± 3.7	67.3 ± 1.8	68.2 ± 1.1	81.0 ± 1.4	73.5 ± 0.1	65.4 ± 0.1	71.6
GEM [12]	71.8 ± 0.6	89.7 ± 2.0	77.0 ± 1.5	78.0 ± 1.4	84.9 ± 1.1	78.2 ± 0.3	67.2 ± 0.6	78.1
KANO [13]	69.9 ± 1.9	90.7 ± 2.2	74.7 ± 2.0	75.7 ± 0.3	82.7 ± 0.9	75.8 ± 0.5	60.2 ± 1.4	75.7
KPGT [26]	71.4 ± 0.7	88.8 ± 2.9	75.7 ± 1.4	77.9 ± 1.2	81.8 ± 2.7	78.5 ± 0.5	64.7 ± 1.0	77.0
MoLFormer [38]	70.9 ± 1.0	91.1 ± 0.9	80.5 ± 1.5	76.7 ± 0.4	83.6 ± 1.1	77.3 ± 0.4	64.9 ± 0.7	77.8
Uni-Mol [63]	72.9 ± 0.6	91.9 ± 1.8	82.1 ± 1.3	80.8 ± 0.3	85.7 ± 0.2	79.6 ± 0.5	65.9 ± 1.3	79.8
Ours _{GIN}	74.1 ± 0.6	95.7 ± 1.2	81.2 ± 0.5	79.8 ± 0.3	85.0 ± 1.1	77.5 ± 0.3	66.7 ± 0.8	80.1
Ours _{GPS}	73.6 ± 0.7	95.1 ± 0.5	81.5 ± 0.8	80.2 ± 0.5	86.1 ± 1.3	79.0 ± 0.6	68.7 ± 0.2	80.6

More importantly, it outperforms most baselines in MoleculeACE by a large margin, while most of the other pre-training methods fail to surpass a simple MLP with fingerprints. The image below (Figure 2a in main text) shows the average model performance across 30 datasets in MoleculeACE. One of the reasons behind KPGT’s good performance could be because it leverages additional fingerprint and molecule descriptors as inputs. Also, we realize that KPGT has already seen over 99% of testing molecules during pre-training, while our method only has seen less than 5%. Even though our method falls slightly behind KPGT, there is still potential room for improvement with a larger pre-training dataset, not to mention that our model is orders of magnitude smaller than KPGT. This suggests that other baselines may either rely more on surface-level molecular features even after pre-training or are more susceptible to knowledge forgetting during fine-tuning. As a result, they struggle with challenging problems that require a subtle understanding of chemical knowledge. The importance of being able to solve activity cliffs within any molecular property prediction task is also highlighted by other works [2, 3].

2. Response to “...molecules might lose their characteristics by molecule perturbation. The learning objectives employed in the proposed method, such as molecule contrastive distancing, scaffold contrastive distancing, and context prediction, all stem from this molecule perturbation concept. Therefore, a contradiction exists between the motivation and methodology.”

There is no contradiction between the motivation and methodology. As mentioned in Introduction, the molecule perturbations that cause the change in molecular characteristics are atom addition/deletion and bond addition/deletion (imagine removing a bond from a ring), which are common graph perturbation strategies. To make the matter worse, these perturbations can violate certain chemical rules and molecule validity, such as incorrect valency. The molecule perturbations we used are subgraph masking and scaffold-invariant molecule perturbation. Subgraph masking only masks out the node-level and edge-level attributes of a subgraph, while retaining the topological structure. This approach is more similar to the dropout strategies for machine learning models. The scaffold-invariant molecule perturbation also guarantees chemical validity, since the perturbations come from the allowable fragment replacement using cheminformatics tools. We also restrict the maximum number of changed atoms to avoid drastic shifts in molecule characteristics.

3. Response to “...However, Figure S5 reveals a consistent tendency for the proposed method to rapidly overfit the training set, contradicting the claimed structure-oriented feature...”

We argue that the performance curve in Figure S6 (Figure S5 in the original submission) indicates the fast convergence rate rather than overfitting. The solid line represents validation performance during fine-tuning, where our method exhibits relatively stable performance (almost a flat line) throughout the training process. It indicates that our method quickly reaches its performance bottleneck on the current dataset, while overfitting would manifest as a clear drop in validation performance. In contrast, the validation curves of other baseline methods exhibit varying degrees of oscillation.

4. Response to “...the comparison in Figure 2 may not be equitable...”

As mentioned in the main text, the representation used for the visualization is the embedding from the readout layer of the graph encoder (e.g., GNN). Then, an additional MLP is used with the representation for property prediction. This is true for all methods in Figure 3 (i.e., Figure 2 in the initial submission). We also argue that the structure-oriented molecular representation is not because of the fixed prompt-guided aggregation. Same as the other methods, the tunable components in our model during fine-tuning include a learnable graph encoder module and a learnable MLP as the prediction head, while the task-agnostic readout layers (e.g., mean pooling of other methods vs our prompt-guided aggregation) are fixed. Therefore, we believe the comparisons are equitable. In terms of the representation from the readout layer, the only difference between using prompt-guided aggregation and mean pooling is one additional linear projection of the node representation (which involves 90,300 parameters compared to 361,200 parameters in encoder), plus the weighted sum. One of the reasons we keep the

aggregation fixed is to prevent everything from collapsing into a single mode, meaning that the outcomes from the three channels become highly similar during fine-tuning.

5. Response to “...*the cliff-noncliff distance ratio should indeed always be positive...*”

Thank you for pointing out this typo. We corrected the statement into “the distance ratio is always above one” . We also included more details about how we calculate the ratio in the updated manuscript.

To reviewer #3:

Thank you for your acknowledgement of our work and for providing detailed and constructive feedback.

1. Response to the general comment of adding additional analysis on the presence of activity cliffs using metrics like SARI index:

Thank you for pointing out the SARI index for examining the discontinuity of the property of interest over the chemical space. In this work, the roughness index (ROGI) we adopt also measures the discontinuity of chemical space (i.e., roughness of the structure-property landscape) and has shown to have a strong correlation with SARI. The main reason we choose ROGI over SARI is because it can generalize to any molecular representations (including fingerprint, descriptors, and neural representation), whereas SARI is tailored specifically for molecule fingerprint. It means that we can use ROGI to not only measure the conventional chemical space discontinuity, but also the potential of different representations in handling the discontinuity. To clarify this for future readers, we included a detailed explanation of the roughness index in the Methods section. We also included an additional experiment (Figure 2b in main text) that shows the correlation between the ROGI value and the model performance. Since ROGI value is a global metric that analyzes the overall chemical space, it does not account for any distribution shift between chemical spaces. In other words, its correlation with the model’s performance on the testing set would be easily obscured by designated data splits, including the stratified splits in MoleculeACE. Therefore, this experiment is performed under the random data split, following the work in [4]. The relationship between the dataset discontinuity (measured by roughness index) and the model performance is shown by the plot below. Compared to the MLP with fingerprint and KPGT (one of the best models in MoleculeACE), our method is shown to be more robust against tasks with

high chemical space discontinuity. The plot also indicates that our method performs better under data scarcity (the size of dots represents the size of datasets).

2. Response to the potency measure in Figure 5 (Figure 4 in the initial submission)

The molar mass units used for potency measure is in nanomolar (nM). In the figure, the value that got normalized is the atom contributions explained by the GNNExplainer. The potency values are labeled on top of the molecule. We apologize for any misunderstandings.

3. Response to the concern of model baselines in MoleculeACE

In the initial submission, we were trying to make the comparison as fair as possible for MoleculeACE, since our method shares the same pre-training dataset, GNN architecture and model hyperparameters as MolCLR and GraphLoG. We have now included 11 baseline methods for MoleculeACE in our latest version. We choose MolFormer as the strong baseline in sequence modeling. It is also the latest work that performs the large-scale SMILES sequence pre-training using the Transformer architecture. In addition, we include Uni-Mol and GEM as strong baseline in methods which leverage 3D geometric information. We also include the performance of MLP with fingerprint, KANO, and KPGT in MoleculeACE. Below is the updated plot, where the x-axis, along with the dots' size, show the model size, and the y-axis shows the average performance in R-squared. The full table is included in Supplementary Information. Our method shows the second

best performance, while most of the other baselines fail to surpass the fingerprint method. One of the reasons behind KPGT's good performance could be because it leverages additional fingerprint and molecule descriptors as inputs. Also, we realize that KPGT has already seen over 99% of testing molecules in pre-training, while our method only has seen less than 5%. Even though our method falls slightly behind KPGT, there is still potential room for improvement with a larger pre-training dataset, not to mention that our model is orders of magnitude smaller than KPGT.

To reviewer #4:

Thank you for your constructive feedback. We have included more analysis in terms of the prompt-guided aggregation and the prompt weight optimization in Supplementary Information.

1. Response to adding more analysis on the context-dependent molecular representations

Thank you for your valuable suggestions. We have performed more analysis on the node aggregation patterns for each channel, and how information from different channels corresponds to knowledge required for different tasks.

We examine the prompt-guided node aggregation patterns on several molecules outside of the pre-training dataset and the patterns are aligned with our expectation. Figure S8 below is the aggregation visualization of a typical example. We highlighted the

Bemis-murcko scaffold and some of the functional groups within the sample molecule. The node attention #1.1 (i.e., attention score from the first attention head within the aggregation module of the first MCD channel) efficiently spans across all atoms, while the node attention #2.1 (i.e., attention score corresponding to the second SCD channel) only captures the atoms belonging to the scaffold. We then show the attention score from all attention heads from the aggregation module for the CP channel. In our experiment, the aggregation attention has 4 heads in total. The atom attention within each head is a lot messier than before. One of the reasons is because CP is not only doing functional group composition prediction, but also masked subgraph prediction. Also, it is impossible to have a one-to-one mapping from attention heads to functional groups. However, we could still observe some local patterns captured by the attention. For example, the node attention #3.4 seems to capture the bicyclic compound, the carbonyl group, as well as the thioether of the molecule.

Even though the prompt-guided node aggregations are fixed during fine-tuning, the graph encoder (e.g, GNN) is still tunable. As the reviewer suggested, we investigate whether the aggregation patterns hold during fine-tuning, especially for the first two channels. Surprisingly, we realize that there is only a slight shift in aggregation patterns. As shown in Figure S9, aggregation patterns from the first two channels are visualized at epoch 20, 50, and 100 when fine-tuning the BBBP dataset. During fine-tuning, the score distribution of attention #1.1 changes slightly, but it continues to span across all atoms. For attention #2.1, the non-scaffold atoms begin to receive increased attention. However, their contributions to the aggregation remain lower than those of the scaffold atoms.

Figure S8: **Case study of prompt-guided aggregation.** Visualization of atom aggregation attention on molecule C(C2C1C(NC(N1)=O)CS2)CCCC(O)=O, along with its scaffold and functional groups. Six different attention distributions are shown from the prompt-guided aggregation. Darker color means higher atom importance.

Figure S9: **Case study of prompt-guided aggregation during fine-tuning.** Visualization of aggregation attention on molecule C(C2C1C(NC(N1)=O)CS2)CCCC(O)=O when fine-tuning BBBP at epoch 20, 50, and 100. Darker color means higher atom importance.

We further conduct a comparison study on four datasets in MoleculeACE to analyze how representations from different channels affect the downstream performance. For starters, we compute the normalized correlation between the differences in molecular feature (molecule fingerprint, scaffold fingerprint, and functional group descriptor) and the differences in potency label. It serves as the QSPR metric to approximate the corresponding knowledge of the task. Table S1 below shows the ablation performance with different channel activations. By examining the columns where only one channel is activated, we observe that the model performance is correlated with the QSPR value. However, when multiple channels are activated, the correlation becomes less obvious. It

seems that incorporating channels with high corresponding QSPR values do not necessarily lead to improved performance. One possible reason is the feature redundancy when combining the two channels, which degrades the model performance. More experimental details and discussion of the results are provided in the Supplementary Information.

Table S1: **Channel activation ablation.** The performance comparison on CHEMBL236_Ki, CHEMBL1871_Ki, CHEMBL228_Ki, and CHEMBL237_Ki datasets when enabling different channels. QSPR stands for the quantitative structure-property relationship metric we used. We manually assign the prompt weights to the channels regarding their activation.

Dataset	QSPR	Channel	Activation						
CHEMBL236_Ki	0.423	MCD	✓			✓		✓	✓
	0.374	SCD		✓		✓	✓		✓
	0.203	CP			✓		✓	✓	✓
	Performance		0.765	0.758	0.756	0.771	0.764	0.776	0.772
CHEMBL1871_Ki	0.391	MCD	✓			✓		✓	✓
	0.272	SCD		✓		✓	✓		✓
	0.338	CP			✓		✓	✓	✓
	Performance		0.523	0.469	0.541	0.529	0.515	0.526	0.518
CHEMBL228_Ki	0.402	MCD	✓			✓		✓	✓
	0.266	SCD		✓		✓	✓		✓
	0.332	CP			✓		✓	✓	✓
	Performance		0.667	0.644	0.662	0.644	0.667	0.667	0.662
CHEMBL237_Ki	0.461	MCD	✓			✓		✓	✓
	0.501	SCD		✓		✓	✓		✓
	0.039	CP			✓		✓	✓	✓
	Performance		0.769	0.772	0.762	0.749	0.765	0.756	0.757

2. Response to the discussion of local optimum of the framework

During the experiments, we realize that the prompt-tuning module may end up prioritizing different channels with different random seeds, which potentially leads to the local optimum. In other words, it means that there might be different ways of combining the knowledge to solve the tasks from the machine learning perspective. This is also consistent with the ablation result above (e.g., model performance reaches 0.667 in CHEMBL228_Ki dataset under three different setups). Therefore, we choose to use roughness index (ROGI) to guide the initialization of the prompt weight as it introduces more insights into the prompt weights from chemoinformatics. However, there is still room for improvement. As we mentioned above, QSPR metrics focus on analyzing the discontinuity of the chemical space (i.e., roughness of the molecular property landscape), but do not consider the distribution shift between chemical spaces. In other words, representation with the smoothest molecular property landscape (i.e., lowest ROGI value) may not lead to the best performance on the testing set created by designated splits (e.g.,

scaffold split in MoleculeNet and stratified split in MoleculeACE). We include more discussion in the Discussion section and leave the investigation of a better prompt-tuning initialization to the future work.

3. Response to “the arrow related to the cliff-noncliff distance ratio seems insignificant”

We admit that the length difference between the two arrows is relatively less obvious in Figure 3 (i.e., Figure 2 in the original submission). Ideally, these data points should be close in space if the structural information is properly encoded within the representation, resulting in a very short distance. The arrow shown in the plot is treated as an example visualization of what the distance looks like. To better illustrate these examples, we update the plot and create a zoom-in region of the arrows. When we said "the red arrow is always larger than the blue arrow", we were talking about the exact example across training epochs, but not all the matched molecular pairs (MMPs). In terms of the actual performance, it is true that not all the cliff MMPs have distance longer than that of the non-cliff MMPs. This is the reason why we report and compare the average distance ratio across methods. We apologize for not making this clear in the text, and we change the wording to “the red arrow is longer than the blue arrow across fine-tuning epochs”.

4. Response to “Is the method applicable to 1D and 3D molecular data?”

In our updated manuscript, we incorporate more baselines into the MoleculeACE benchmark, but fail to see any clear advantages of using either 1D or 3D representations alone (i.e., the performance comparison between Uni-Mol, MoLFormer, and other graph models on MoleculeACE). However, we do believe that it is promising to incorporate different input representations into our framework, such that the model will not only leverage knowledge from different structural levels, but also knowledge from different dimensions (1d sequence, 2d topology, and 3d geometry). We provide a thorough discussion in ways of extending our framework with different representations in Supplementary Information.

In terms of using 1d (SMILES sequence) or 3d (geometric information) representation alone, the framework adjustment should be straightforward. Since scaffold invariant perturbation essentially creates a new molecule, there would be no issue of changing the input representations. The only tricky part is the definition of subgraph within molecule contrastive distancing and masked subgraph prediction. Since a subsequence of the SMILES string does not necessarily correspond to a valid chemical structure, we would need to tag the subgraph from 2d molecular graphs and map it back to the SMILES string (e.g., assigning special tags to the SMILES token, or changing the token symbol). For 3d

representations, the graph definition may vary. However, since the subgraph masking is only performed at the attribute level, it is compatible with 3d graphs as well.

Another interesting extension would be to incorporate different input representations into the framework, treating each representation as a different modality. Essentially, there would be multiple molecule encoders, such as a geometry encoder for 3D representation and a graph encoder for 2D representation. While each encoder processes the molecule in its own unique way, we could enable communication between encoders so that the final encoded representation integrates knowledge from all dimensions. This joint learning could benefit the learning of individual encoders. Moreover, cross-modality alignment tasks could be used to explicitly enforce the channel alignment. This multimodal framework may become more transferable to tasks that require different levels of knowledge (e.g., structure composition versus conformation information).

- [1] Sun, R., Dai, H. & Yu, A. W. Does GNN Pretraining Help Molecular Representation? *36th Conference on Neural Information Processing Systems*. (2022).
- [2] Deng, J. *et al.* A systematic study of key elements underlying molecular property prediction. *Nat Commun* **14**, 6395 (2023).
- [3] Van Tilborg, D., Alenicheva, A. & Grisoni, F. Exposing the Limitations of Molecular Machine Learning with Activity Cliffs. *J. Chem. Inf. Model.* **62**, 5938–5951 (2022).1.
- [4] Aldeghi, M. *et al.* Roughness of Molecular Property Landscapes and Its Impact on Modellability. *J. Chem. Inf. Model.* **62**, 4660–4671 (2022).

REVIEWER COMMENTS

Reviewer #2 (Remarks to the Author):

The authors have provided a more comprehensive evaluation of the proposed method's performance. However, I feel that some of my previous comments have not been fully addressed. There are a few remaining issues in the manuscript that I believe need further attention. The specific points are as follows:

1. The authors did not provide point-by-point responses to my comments, and some of my comments were ignored. Additionally, highlighting the revised content in the manuscript would greatly assist in identifying the new additions.
2. The results presented in Figure 2b are somewhat confusing. According to Table S2, the GPS version of the proposed method did not achieve an R-squared score higher than 0.9. However, in Figure 2b, one data point representing this model exceeds 0.9. There appear to be many inconsistencies like this. Please review the results to ensure consistency.
3. The ablation study provided is a good start, particularly with the channel activation results in Table S1. However, the proposed model contains many components, and a more comprehensive ablation study covering each introduced component would provide a clearer understanding of their specific contributions.

Reviewer #3 (Remarks to the Author):

After the first round of revisions and corrections, I certify that the article is suitable for acceptance in Nature Communications.

Reviewer #4 (Remarks to the Author):

I commend the authors for their diligent efforts in enhancing their manuscript. The revised paper is well-organized and clearly articulated. The experiments are now more comprehensive, with thorough analysis that broadens its appeal to the audience of Nature Communications. Additionally, I am pleased to note that most of my concerns have been satisfactorily addressed in the rebuttal and supplementary information. While I am inclined to recommend the publication of this paper, there are a few additional questions that we would like to discuss with the authors:

1. Analysis of context-dependent representations and the prompt select module.

It is interesting to see that the model performance is correlated to the QSPR measure of the molecular knowledge required for different tasks. The feature redundancy issue in combining multiple channels seems an inspiring topic for future exploration. Hence, I'm curious about the following questions:

(1.1) Is the optimized prompt weights correlate with the QSPR metrics?

(1.2) Is the design prompt selection module optimal for combining representations from different channels? Compared with the learnable linear combination proposed in this work, will the direct

concatenation of multi-channel representations or a mixture-of-expert (MoE) module lead to improved results (as indicated in Table S1, by selecting a single channel)?

2. The cliff-noncliff distance ratio

While the definitions and analysis of the cliff-noncliff distance ratio has been clearly stated, I believe that visualizing the cliff-noncliff distance ratio for MolCLR and the proposed model will be beneficial for answering the following questions of the activity cliff issue:

(2.1) Does the model pull the noncliff MMPs closer, push the cliff MMPs further, or both?

(2.2) Is it possible to investigate the contribution of each part of the pre-training design in addressing the nuances of chemical knowledge?

3. Performance comparison on MoleculeNet

The incorporation of more baselines that involve 1d and 3d molecular data is credited. However, two concerns should be further addressed:

(3.1) In lines 229-231, the authors stated that 'there are no clear advantages to using either 1D sequence or 3D geometry for binding potency prediction', which may lead to misunderstanding, as 3D geometry is the most fundamental component for binding potency. I think the authors would like to claim that "pre-trained models using 1D sequence or 3D geometry have no clear advantages for binding potency prediction".

(3.2) The performance of KANO replicated by authors shows discrepancies with the original paper (due to different splits). Some reimplementations details could be discussed.

To all reviewers:

We would like to thank all reviewers once again for their time and efforts in reviewing our updated manuscripts, as well as the constructive feedback and valuable suggestions. For your convenience, we highlight the updated contents in the manuscript from the last time in **red**, and the main modification from the previous round in **blue**. We hope that this revision addresses all remaining concerns.

To reviewer #2:

Thank you for your constructive feedback. We have included more ablation studies in terms of both the pre-training components and the fine-tuning strategies in our updated manuscripts. We also highlighted the updated contents for your convenience. Thank you for your suggestion.

1. In response to *“The authors did not provide point-by-point responses to my comments, and some of my comments were ignored. Additionally, highlighting the revised content in the manuscript would greatly assist in identifying the new additions.”*

We apologize for not fully addressing all of your concerns in our previous response. For your convenience, we reformat our last response into a point-by-point format and submitted it as an additional file. We hope this will make our response easier to follow. Additionally, we have highlighted the updated content in the manuscript in **red**, with the main modifications from the previous round highlighted in **blue**. Thank you for your valuable suggestions.

2. In response to *“The results presented in Figure 2b are somewhat confusing. According to Table S2, the GPS version of the proposed method did not achieve an R-squared score higher than 0.9. However, in Figure 2b, one data point representing this model exceeds 0.9. There appear to be many inconsistencies like this. Please review the results to ensure consistency.”*

We apologize for any confusion regarding the results. The experiments shown in Figures 2a and 2b were conducted differently. Figure 2a and Table S2 present the results based on the stratified splits proposed in MoleculeACE, while Figure 2b is evaluated using random splits. The reasoning behind is as follows:

As described in the main text, Figure 2b aims to demonstrate the relationship between the degree of structure-property discontinuity (e.g., the amount of activity cliffs) within each dataset in MoleculeACE and the performance of different methods in handling such discontinuity. This discontinuity is often measured by quantitative structure-property relationship (QSPR) metrics. In this work, we use the roughness index (ROGI) [1] to quantify this aspect. A lower ROGI value indicates a smoother structure-property landscape, and the more likely for the machine learning model to perform well. However, this relationship applies primarily to random splits because QSPR metrics evaluate the entire chemical space without considering any distributional shifts between the training, validation, and test sets. In other words, if we adopt the same stratified splits as in Figure 2a, the correlation between landscape roughness and model performance on the test set can be easily obscured by the specific data splits used. Below is an illustration of the correlation observed under random splits compared to MoleculeACE's stratified splits. The experiment is done using MLP with molecular fingerprints on the 30 datasets in MoleculeACE.

To avoid future confusion, we have modified the captions of all performance tables and figures to include the detailed settings of the experiments. We have also included the full performance table of Figure 2b in the supplementary information. In general, all experiments that involve the QSPR analysis are done with the random splits. Otherwise, the experiments are performed using the stratified splits in the MoleculeACE benchmark.

3. In response to *“The ablation study provided is a good start, particularly with the channel activation results in Table S1. However, the proposed model contains many components, and a more comprehensive ablation study covering each introduced component would provide a clearer understanding of their specific contributions.”*

We include a more comprehensive ablation study that covers the pre-training components, fine-tuning strategies, and the relationship between optimized prompt weights and the knowledge required for solving downstream tasks.

More specifically, we evaluate the effectiveness of the adaptive margin loss, regularization techniques, and the multi-channel learning framework during pre-training. We follow the same pretrain-finetune workflow as in the main experiments across 6 different pre-training settings: 1. The full pre-train setting, corresponding to the best model configuration described in the main text. 2. Without any regularizations. 3. Without intra-channel regularization on node aggregation patterns. 4. Without inter-channel regularization on channel alignment. 5. Replacing the adaptive margin loss with the conventional margin loss. 6. Replacing the multi-channel learning with the conventional multi-task learning, where there is only one aggregation channel that learns all the pre-trained tasks. All settings are pre-trained on ZINC15 using the GIN backbone for 40 epochs and evaluated on the MoleculeACE benchmark.

Figure S10 (also shown below) presents the performance comparison of the average test R-squared values across the 30 datasets in MoleculeACE. The error bars represent the average standard deviation from three runs across the 30 datasets. As indicated by the plot, removing either the regularization components, or the adaptive margin loss results in a similar degree of performance degradation. Notably, the regularization of prompt-guided node aggregation (i.e., intra-channel regularization) appears to play a more critical role than channel alignment (i.e., inter-channel regularization). The largest performance drop is observed in the multi-task learning setting. It is important to note that the only difference between this setting and the multi-channel learning setting is that the latter framework learns the same set of tasks in separate channels. This highlights the **effectiveness** of multi-channel learning, as it decomposes the pre-trained tasks based on different aspects of chemical knowledge, allowing for the combination of pre-trained knowledge in a task-specific manner. We have included the full performance table in Table S4.

Additionally, we examine three different strategies for leveraging the multi-channel representations during fine-tuning: 1. Concat, where the channel-wise representations are simply concatenated together for downstream prediction. 2. Learnable prompt weight (PW), where task-specific prompt weights are initialized via the guidance of ROGI value, and learned during fine-tuning. This corresponds to the setting of our best model. 3. Mixture-of-expert (MoE) alike learnable weights, which is both task-specific and sample-specific. This is archived by a gating network that takes the graph representation obtained from mean pooling and outputs the set of importance weights for each expert (i.e., channel). For the latter two methods, a weighted sum is applied to the channel-wise representations to get the composed representation for prediction. We also explore how channel sparsity affects performance. The sparsity level is controlled by dividing the channel logits by a temperature t : $weights = Softmax(logits/t)$. We examine the cases where the temperature $t=1/0.7/0.3$. We utilize the same pre-trained checkpoint of GIN as the Figure 2a for this ablation. More formulation and implementation details are included in the ablation section of Supplementary Information.

As shown in Figure S11 (also shown below), Sparse PW with $t = 0.7$ achieves the best performance, corresponding to the best model setting described in the main text. The Concat method, which evenly utilizes all available information, performs slightly worse than PW. One possible explanation for MoE's poor performance is its tendency to overfit due to its added dimensionality and complexity. Additionally, unlike the learnable PW, MoE does not incorporate prior information from the ROGI measure, highlighting the **advantages** of ROGI-guided initialization. Another interesting observation is the tradeoff between information sparsity and completeness. For both PW and MoE, performance improves when the temperature decreases from $t = 1$ to $t = 0.7$ but drops when further reduced to $t = 0.3$. This suggests that some degree of channel selectivity is beneficial. However, as the temperature continues to

decrease, the model will tend to rely on a single channel for predictions, ignoring information from other channels. This approach proves to be less effective than leveraging multiple channels. We have included the full performance table in Table S5.

Last but not least, we further examine the relationship between the optimized prompt weights and the knowledge required for solving the tasks, expanding on the existing case study about channel activation. Specifically, we first retrieve the optimized prompt weights (PW) from the best validation model after fine-tuning on each of the 30 datasets in MoleculeACE using the random split. We also collect the same QSPR measure presented in Table S1, which is the normalized correlation between representation difference and label difference, for all datasets. We then perform a principal component analysis (PCA) on these two sets of normalized vectors and visualize their relationship in a scatter plot using the computed first principal components.

As shown in Figure S12 (also shown below), we observe **some correlations** between the optimized prompt weights (PW) and QSPR measures. However, the alignment is far from perfect. One possible reason could be the feature redundancy hidden within the channel-wise representations, which may influence the model's decision. Additionally, due to the black-box nature of machine learning models, the problem may be solved in ways that differ from human interpretation. This suggests that hidden information might be processed differently by the model, which requires further investigation. Moreover, it is important to note that **large-scale pre-training does not necessarily endow the model with perfect capability in solving the pre-train tasks**. Learning multiple tasks simultaneously can still pose challenges, even within the multi-channel learning framework. This indicates that there may be information gaps between the learned channel-wise representations and the actual chemical knowledge they are meant to capture.

An intriguing direction for future work would be to analyze how the model’s actual capabilities, as learned during pre-training, influence fine-tuning performance. Additionally, understanding how to appropriately align each pre-training component with performance improvements on different fine-tuning tasks could provide valuable insights for optimizing such models.

To reviewer #3:

Thank you for your thorough review and positive feedback! We appreciate your efforts and are glad that our revisions have addressed all of your concerns.

To reviewer #4:

Thank you for your acknowledgement and the interesting discussion questions. We've added further analysis to address each one of them.

1. In response to more discussion in analyzing the context-dependent representations and the prompt selection module.

1.1. *“Does the optimized prompt weight correlate with the QSPR metrics?”*

Thank you for pointing this out. This is an interesting question, and we have included a follow-up analysis in the Supplementary Information. The goal of this

analysis is to quantitatively measure the correlation between the optimized prompt weights and the QSPR metric used in Table S1. To begin, we retrieve the optimized prompt weights (PW) from the best validation model after fine-tuning on each of the 30 datasets in MoleculeACE using the random split. We then collect the same QSPR measure from Table S1 (i.e., the normalized correlation between representation difference and label difference) for all datasets. Finally, we conduct a principal component analysis (PCA) on these two sets of normalized vectors and visualize their relationship in a scatter plot using the computed first components.

As shown in Figure S12 (also shown below), we observe **some correlations** **between the optimized prompt weights (PW) and QSPR measures**. However, the alignment is far from perfect. One possible reason could be the feature redundancy hidden within the channel-wise representations, which may influence the model's decision. Additionally, due to the black-box nature of machine learning models, the problem may be solved in ways that differ from human interpretation. This suggests that hidden information might be processed differently by the model, which requires further investigation. Moreover, it is important to note that **large-scale pre-training does not necessarily endow the model with perfect capability in solving the pre-train tasks**. Learning multiple tasks simultaneously can still pose challenges, even within the multi-channel learning framework. This indicates that there may be information gaps between the learned channel-wise representations and the actual chemical knowledge they are meant to capture. An intriguing direction for future work would be to analyze how the model's actual capabilities, as learned during pre-training, influence fine-tuning performance.

1.2. *“Is the design prompt selection module optimal for combining representations from different channels? Compared with the learnable linear combination proposed in this work, will the direct concatenation of multi-channel representations or a mixture-of-expert (MoE) module lead to improved results (as indicated in Table S1, by selecting a single channel)?”*

To address this question, we include an ablation study that compares the effects of different fine-tuning strategies on downstream performance, including concatenation (Concat), the learnable prompt weights (PW) used in our main experiment, and a Mixture-of-Experts (MoE)–like prompt selection module. The MoE method is implemented through a gating network that takes in the graph representation obtained from mean pooling and outputs the importance weight for each channel. The primary difference between PW and MoE, aside from the added complexity, is that PW is task-specific but sample-agnostic, while MoE is both task-specific and sample-specific. This means that channel importance can vary across different molecules. Additionally, we experiment with temperature settings of $t=1/0.7/0.3$, to control the sharpness of the prompt distribution (i.e., the sparsity of the selected channels).

As shown in Figure S11 (also shown below), Sparse PW with $t = 0.7$ achieves the best performance, corresponding to the best model setting described in the main text. The Concat method, which evenly utilizes all available information, performs slightly worse than PW. One possible explanation for MoE’s poor performance is its tendency to overfit due to its added complexity. Additionally, unlike the learnable PW, MoE does not incorporate prior information from the ROGI measure, highlighting the **advantages** of ROGI-guided initialization. Another interesting observation is the tradeoff between information sparsity and completeness. For both PW and MoE, performance improves when the temperature decreases from $t = 1$ to $t = 0.7$ but drops when further reduced to $t = 0.3$. This suggests that some degree of channel selectivity is beneficial. However, as the temperature continues to decrease, the model will tend to rely on a single channel for predictions, ignoring information from other channels. This approach proves to be less effective than leveraging multiple channels.

2. In response to a more fine-grained visualization for the cliff-noncliff distance ratio:

2.1. *“Does the model pull the noncliff MMPs closer, push the cliff MMPs further, or both?”*

We visualize the detailed distance histograms of non-cliff MMPs and cliff MMPs from the fine-tuning experiment in Figure 3, as shown in Figure S13 (also presented below). Each row represents one method, while each column corresponds to a training step. For clarity, the distances are normalized between 0 and 1, and the x-axis range is standardized across all subplots to ensure consistency in the visual comparison.

Unfortunately, we do not observe that the non-cliff MMPs are pulled closer during fine-tuning. For both GraphLoG and our method, non-cliff and cliff MMPs

are pushed further apart along the training process. This is understandable, as the learned representations become more label-oriented rather than structure-oriented during fine-tuning, leading to some loss of structural information. However, it remains important for the model to maintain distance differences between cliff pairs and non-cliff pairs for understanding of activity cliffs throughout fine-tuning, which is particularly true for our method. In contrast, MolCLR exhibits more oscillatory behavior, with less consistent distance distribution patterns during fine-tuning. This corresponds to the drastic shifts in representation space shown in Figure 3.

2.2. *“Is it possible to investigate the contribution of each part of the pre-training design in addressing the nuances of chemical knowledge?”*

As discussed above, aligning the pre-training components with the model's capability to understand the nuances of chemical knowledge could be challenging. One difficulty lies in the actual capability of the pre-trained models. Large-scale pre-training does not guarantee that the model will be perfectly capable of solving the pre-training tasks, and learning multiple tasks simultaneously can still be difficult within a multi-channel learning framework. This suggests that there may be gaps between the learned representations and actual chemical knowledge. Learning conflicts may also arise in multi-task learning, which needs further investigation. One intriguing direction for future research is to analyze how the capability of individual pre-trained tasks impacts fine-tuning performance and to explore ways to improve joint learning more efficiently.

Even if the pre-trained model perfectly understands the pre-trained knowledge, it remains challenging to assess which aspects of the pre-training design correspond to the nuances of chemical knowledge in downstream applications. To comprehensively evaluate the contribution of each pre-training component to fine-tuning performance, an ideal experimental setup would involve pre-training

the model under all possible combinations of pre-training components, which may not be accessible. What is more, since fine-tuning performance can be influenced by many factors beyond pre-training (e.g., dataset size, distribution shifts among data subsets, data balancing/diversity within the dataset), it is difficult to determine whether good performance is due to the learned chemical knowledge embedded within the pre-training components or whether the learned representation space is simply more resilient to these factors. In summary, while we were able to perform correlation analysis at certain levels, it is challenging to establish a definitive causal relationship between the contribution of each pre-training component and the model's ability to capture the nuances of chemical knowledge during fine-tuning.

3. In response to *“Performance comparison on MoleculeNet”*

3.1. *“In lines 229-231, the authors stated that 'there are no clear advantages to using either 1D sequence or 3D geometry for binding potency prediction', which may lead to misunderstanding, as 3D geometry is the most fundamental component for binding potency. I think the authors would like to claim that 'pre-trained models using 1D sequence or 3D geometry have no clear advantages for binding potency prediction'.”*

Thank you for pointing out the inappropriate statement. We have revised it to: “There are no clear advantages of molecular representation learning using either 1D sequence or 3D geometry.”

3.2. *“The performance of KANO replicated by authors shows discrepancies with the original paper (due to different splits). Some reimplementations details could be discussed.”*

We have added more details in the Methods section to explain the difference between the two data splits (deterministic scaffold split and balanced scaffold split) and the reimplementations details. Both splits aim to hold out a set of molecules with scaffolds that are not seen during training. The deterministic scaffold split prioritizes maintaining the specified split ratio, while the balanced scaffold split prioritizes balancing scaffold frequency across the training, validation, and test sets. We use the deterministic scaffold split in our experiments primarily because most of the baselines also employ it. All other aspects, including hyperparameter choices for both the models and the training

setup, remain consistent with their default configurations for replicating the results.

[1] Aldeghi, M. *et al.* Roughness of Molecular Property Landscapes and Its Impact on Modellability. *J. Chem. Inf. Model.* **62**, 4660–4671 (2022).

To reviewer #2:

We thank you for your valuable time and efforts in reviewing our manuscripts. For your convenience, we reformat the previous response into a point-by-point format. We hope that your concerns could be addressed by the following response.

1. In response to *“The major components of the proposed framework are adopted from previous studies, such as the objectives (molecule contrastive learning, motif prediction, and masked subgraph prediction) and the prompt-based aggregation module. Moreover, the prediction performance of the proposed framework is also less than ideal, as it achieved comparable or worse performance on 4 out of 7 datasets from the MoleculeNet benchmark. Therefore, this work seems to provide limited contributions to this field.”*

The main goal of this work is to build a generic and highly generalizable molecular representation that is not only suitable to learn molecular properties across a diverse range of molecular scaffolds, but also captures the subtle variations across fine-scaled structural distinctions (such as the phenomenon of activity cliffs) among highly similar molecules. One major innovation of the proposed method is its flexibility to combine molecule representations at different levels (overall structure, scaffold, functional group) according to the needs of downstream prediction tasks. The proposed method contrasts with existing methods that focus on a single type of representation, which improves its adaptability to various prediction tasks.

Additionally, as we discuss the issues with existing methods in the Introduction section, we further refine the learning method for each type of molecule representation based on the chemical knowledge. We leverage contrastive learning in our framework since it is well aligned with the common heuristics in chemistry and with our goals. However, our contrastive formulation, built on top of the adaptive triplet loss, is largely different from the previous contrastive methods that utilized the NT-Xent loss (e.g., MolCLR, KANO). We also propose a novel scaffold-invariant molecular perturbation method, which no one has done before. Also, the conventional motif prediction is framed as the multi-label binary classification task, which is shown to be less helpful than other SSL tasks [1]. Therefore, we adjust it into a regression task that predicts the normalized functional group descriptors and improves model learning.

Finally, an important contribution of this work is the development of an effective model for molecular property prediction that naturally handles the issues of

activity cliffs. We argue that it is essential to consider the performance from both benchmarks (i.e. MoleculeNet and MoleculeACE) simultaneously to evaluate the quality of the pre-trained representations. Even though our method does not significantly improve the SOTA performance in MoleculeNet, its average performance, 80.6 in AUC, is **still the highest** among the other strong baselines. The second best method is Uni-Mol, which has an average AUC score of 79.8, but is an order of magnitude larger than our model. Moreover, Uni-Mol cannot competitively handle issues of activity cliffs as illustrated in a later figure below.

Table 1: Fine-tuning results on 7 classification tasks in MoleculeNet **using the scaffold splits**. Average ROC-AUC value is reported, along with the score standard deviation from three independent runs. The first two rows (GIN and GPS) show the results under the no-pretrain setting.

Methods	BBBP	Clintox	MUV	HIV	BACE	Tox21	SIDER	Avg.
#task	1	2	17	1	1	12	27	
GIN [59]	65.8 ± 4.5	58.0 ± 4.4	71.8 ± 2.5	75.3 ± 1.9	70.1 ± 5.4	74.0 ± 0.8	57.3 ± 1.6	67.5
GPS [36]	64.8 ± 3.0	87.2 ± 0.9	69.8 ± 3.8	73.1 ± 3.3	78.0 ± 3.0	74.5 ± 0.6	60.8 ± 0.6	72.6
Hu et. al [22]	70.8 ± 1.5	72.6 ± 1.5	81.3 ± 2.1	79.9 ± 0.7	84.5 ± 0.7	78.7 ± 0.4	62.7 ± 0.8	75.8
GraphLoG [60]	72.3 ± 0.9	74.7 ± 2.2	74.2 ± 1.8	75.4 ± 0.6	82.2 ± 0.9	75.1 ± 0.7	61.2 ± 1.1	73.6
D-SLA [25]	72.6 ± 0.8	80.2 ± 1.5	76.6 ± 0.9	78.6 ± 0.4	83.8 ± 1.0	76.8 ± 0.5	60.2 ± 1.1	75.5
MolCLR [52]	73.5 ± 0.4	90.4 ± 1.7	75.5 ± 1.8	77.6 ± 3.2	83.5 ± 1.8	76.7 ± 2.1	60.7 ± 5.7	76.8
GraphMVP [28]	72.4 ± 1.6	79.1 ± 2.8	77.7 ± 0.6	77.0 ± 1.2	81.2 ± 0.9	75.9 ± 0.5	63.9 ± 1.2	75.3
GROVER [38]	69.5 ± 0.1	76.2 ± 3.7	67.3 ± 1.8	68.2 ± 1.1	81.0 ± 1.4	73.5 ± 0.1	65.4 ± 0.1	71.6
GEM [13]	71.8 ± 0.6	89.7 ± 2.0	77.0 ± 1.5	78.0 ± 1.4	84.9 ± 1.1	78.2 ± 0.3	67.2 ± 0.6	78.1
ImageMol [64]	73.9 ± 0.2	85.1 ± 1.4	82.5 ± 0.8	79.7 ± 0.2	83.9 ± 0.5	77.3 ± 0.1	66.0 ± 0.1	78.3
KANO [14]	69.9 ± 1.9	90.7 ± 2.2	74.7 ± 2.0	75.7 ± 0.3	82.7 ± 0.9	75.8 ± 0.5	60.2 ± 1.4	75.7
KPGT [27]	71.4 ± 0.7	88.8 ± 2.9	75.7 ± 1.4	77.9 ± 1.2	81.8 ± 2.7	78.5 ± 0.5	64.7 ± 1.0	77.0
MoLFormer [39]	70.9 ± 1.0	91.1 ± 0.9	80.5 ± 1.5	76.7 ± 0.4	83.6 ± 1.1	77.3 ± 0.4	64.9 ± 0.7	77.8
Uni-Mol [66]	72.9 ± 0.6	91.9 ± 1.8	82.1 ± 1.3	80.8 ± 0.3	85.7 ± 0.2	79.6 ± 0.5	65.9 ± 1.3	79.8
Ours _{GIN}	74.1 ± 0.6	95.7 ± 1.2	81.2 ± 0.5	79.8 ± 0.3	85.0 ± 1.1	77.5 ± 0.3	66.7 ± 0.8	80.1
Ours _{GPS}	73.6 ± 0.7	95.1 ± 0.5	81.5 ± 0.8	80.2 ± 0.5	86.1 ± 1.3	79.0 ± 0.6	68.7 ± 0.2	80.6

More importantly, it outperforms most baselines in MoleculeACE by a large margin, while most of the other pre-training methods fail to surpass a simple MLP with fingerprint. The image below (Figure 2a in main text) shows the average model performance across 30 datasets in MoleculeACE. One of the reasons behind KPGT’s good performance could be because it leverages additional fingerprint and molecule descriptors as inputs. Also, we realize that KPGT has already seen over 99% of testing molecules during pre-training, while our method only has seen less than 5%. Even though our method falls slightly behind KPGT, there is still potential room for improvement with a larger pre-training dataset, not to mention that our model is orders of magnitude smaller than KPGT. This suggests that other baselines may either rely more on surface-level molecular features even after pre-training or are more susceptible to knowledge forgetting during fine-tuning. As a result, they struggle with challenging problems that

require a subtle understanding of chemical knowledge. The importance of being able to solve activity cliffs within any molecular property prediction task is also highlighted by other works [2, 3].

2. In response to *“As discussed in the Introduction section, molecules might lose their characteristics by molecule perturbation. The learning objectives employed in the proposed method, such as molecule contrastive distancing, scaffold contrastive distancing, and context prediction, all stem from this molecule perturbation concept. Therefore, a contradiction exists between the motivation and methodology. Moreover, I think that this discrepancy is the cause of the observed lack of significant improvement over baseline methods in the proposed approach.”*

There is no contradiction between the motivation and methodology. As mentioned in Introduction, the molecule perturbations that cause the change in molecular characteristics are atom addition/deletion and bond addition/deletion (imagine removing a bond from a ring), which are common graph perturbation strategies. To make the matter worse, these perturbations can violate certain chemical rules and molecule validity, such as incorrect valency. The molecule perturbations we used are subgraph masking and scaffold-invariant molecule perturbation. Subgraph masking only masks out the node-level and edge-level attributes of a subgraph, while retaining the topological structure. This approach is more similar to the dropout strategies for machine learning models. The

scaffold-invariant molecule perturbation also guarantees chemical validity, since the perturbations come from the allowable fragment replacement using cheminformatics tools. We also restrict the maximum number of changed atoms to avoid drastic shifts in molecule characteristics.

3. In response to *“The prediction performance of the proposed method need to be further evaluated. The authors discussed that current self-supervised learning-based methods on molecules perform worse than the molecule fingerprint-based methods in the Introduction section. Therefore, molecule fingerprint-based methods should be included as baseline methods in the computational tests. Moreover, key self-supervised learning baseline methods are missed in the benchmarking tests, including KPGT[1], ImageMol[2], and MolFormer[3]. More baseline methods should also be incorporated into the benchmarking tests on the datasets from MoleculeACE.”*

Thank you for your suggestions. We have now included up to 14 baseline methods to compare with, including KPGT, ImageMol, and MolFormer. These baseline methods are diverse in terms of pre-training strategies, input representations (e.g., SMILES/Graph/3D geometry), model backbones (GNN vs Transformer-based), and model sizes.

4. In response to *“The authors claim that the proposed method prioritizes structure over labels during fine-tuning (line 255, page 8). However, Figure S5 reveals a consistent tendency for the proposed method to rapidly overfit the training set, contradicting the claimed structure-oriented feature. It is unclear whether a multiple-layer perceptron (MLP) is utilized for predictions on top of the pre-trained model. If so, the comparison in Figure 2 may not be equitable. Despite the pre-trained model generating structure-oriented molecular representations (due to the fixed prompt-guided aggregation module during training), the inner layer of the MLP could introduce label-oriented aspects.”*

We argue that the performance curve in Figure S6 (Figure S5 in the original submission) indicates the fast convergence rate rather than overfitting. The solid line represents validation performance during fine-tuning, where our method exhibits relatively stable performance (almost a flat line) throughout the training process. It indicates that our method quickly reaches its performance bottleneck on the current dataset, while overfitting would manifest as a clear drop in validation performance. In contrast, the validation curves of other baseline methods exhibit varying degrees of oscillation.

As mentioned in the main text, the representation used for the visualization is the embedding from the readout layer of the graph encoder (e.g., GNN). Then, an additional MLP is used with the representation for property prediction. This is true for all methods in Figure 3 (i.e., Figure 2 in the initial submission). We also argue that the structure-oriented molecular representation is not because of the fixed prompt-guided aggregation. Same as the other methods, the tunable components in our model during fine-tuning include a learnable graph encoder module and a learnable MLP as the prediction head, while the task-agnostic readout layers (e.g., mean pooling of other methods vs our prompt-guided aggregation) are fixed. Therefore, we believe the comparisons are equitable. In terms of the representation from the readout layer, the only difference between using prompt-guided aggregation and mean pooling is one additional linear projection of the node representation (which involves 90,300 parameters compared to 361,200 parameters in encoder), plus the weighted sum. One of the reasons we keep the aggregation fixed is to prevent everything from collapsing into a single mode, meaning that the outcomes from the three channels become highly similar during fine-tuning.

5. In response to *“Given the incorporation of numerous components in the proposed framework, it is essential to conduct comprehensive ablation studies to assess the necessity of each element.”*

At the latest revision, we have included more ablation study in pre-train components, fine-tune strategies, and how optimized prompt weight and channel

activation align with the required knowledge for solving the downstream tasks. Please see the other response document for more details.

6. In response to *"It is imperative to make the codes and datasets employed in this study publicly available. This accessibility is crucial not only for evaluating the model but also for the broader utility within the scientific community."*

Both the codes and the datasets in this study are publicly available in <https://github.com/yuewan2/MolMCL/tree/main>.

7. In response to *"There are typos in the manuscript. In line 252, "can be better transferedto the target application" should be revised to "can be better transferred to the target application". In line 265, "the red arrow is always larger than the blue arrow" should be revised to "the red arrow is always longer than the blue arrow"."*

Thank you for pointing this out. We changed the wording from "larger" to "longer".

8. In response to *"Additionally, is it correct to state "Our cliff-noncliff distance ratio in the validation set is always positive"? As per my understanding, the cliff-noncliff distance ratio should indeed always be positive."*

Thank you for pointing out this typo. We corrected the statement into "the distance ratio is always above one". We also included more details about how we calculate the ratio in the updated manuscript.

9. In response to *"The arrows in Figure 2 do not effectively convey their intended meaning. Consider using alternative visual representations for clarity."*

We understand that it can be hard to see the distance difference in Figure 2, especially for the cases where matched molecular pairs (MMPs) are often close in space. For better visual illustration, we create a zoom-in region of the arrows in Figure 2. We also include its corresponding detailed distance histogram of the MMPs, along with more discussion in Figure S12 of Supplementary Information.

10. In response to *"The definitions of "bonded results" and the term "+-" should be explicitly provided in the captions of Tables 1 and 2 to enhance clarity."*

We adjust the caption and explain the term “+–” as the standard deviation of three independent runs on each dataset.

[1] Sun, R., Dai, H. & Yu, A. W. Does GNN Pretraining Help Molecular Representation? *36th Conference on Neural Information Processing Systems*. (2022).

REVIEWERS' COMMENTS

Reviewer #2 (Remarks to the Author):

The authors have provided a detailed and scholarly response to my concerns. I believe the manuscript has been significantly improved and is now suitable for acceptance for publication in Nature Communications.

Reviewer #4 (Remarks to the Author):

We thank the authors for their comprehensive experiments and the additional clarifications provided. All of my concerns have been satisfactorily addressed.